# Efficient Submodular Optimization under Noise: Local Search is Robust

**Lingxiao Huang**
Huawei TCS Lab -> Nanjing University
`huanglingxiao1990@126.com`

**Yuyi Wang**
Swiss Federal Institute of Technology
`yuyiwang920@gmail.com`

**Chunxue Yang**
Nanyang Technological University
`chunxue001@e.ntu.edu.sg`

**Huanjian Zhou**
The University of Tokyo
`zhou@ms.k.u-tokyo.ac.jp`

## Abstract

The problem of monotone submodular maximization has been studied extensively due to its wide range of applications. However, there are cases where one can only access the objective function in a distorted or noisy form because of the uncertain nature or the errors involved in the evaluation. This paper considers the problem of constrained monotone submodular maximization with noisy oracles introduced by [11]. For a cardinality constraint, we propose an algorithm achieving a near-optimal $\left(1 - \frac{1}{e} - O(\varepsilon)\right)$-approximation guarantee (for arbitrary $\varepsilon > 0$) with only a polynomial number of queries to the noisy value oracle, which improves the exponential query complexity of [20]. For general matroid constraints, we show the first constant approximation algorithm in the presence of noise. Our main approaches are to design a novel local search framework that can handle the effect of noise and to construct certain smoothing surrogate functions for noise reduction.

## 1 Introduction

Consider the following problems in machine learning and operations research: (1) selecting a set of locations to open up facilities with the goal of maximizing their overall user coverage [15]; (2) reducing the number of features in a machine learning model while retaining as much information as possible [23]; and (3) identifying a small set of seed nodes that can achieve the largest overall influence in a social network [14]. Solving these problems all involves maximizing a monotone *submodular* set function $f : 2^N \mapsto \mathbb{R}$ subject to certain constraints. Intuitively, submodularity captures the property of diminishing returns. For example, a newly opened facility will contribute less to the overall user coverage if we have already opened many facilities and more if we have only opened a few. Although the general problem of monotone submodular maximization subject to a cardinality or general matroid constraint is NP-hard [5], the greedy algorithm, which selects an element with the largest margin at each step, can approximately solve this problem under a cardinality constraint by a factor of $1 - 1/e$, and this approximation ratio is tight [18]. Moreover, a non-oblivious local search algorithm, which iteratively alters one element to improve an auxiliary objective function, is guaranteed to achieve an approximation ratio of $1 - 1/e$ for general matroid constraints [7].

In the literature, the submodular optimization problem usually assumes a *value oracle* to the objective function $f$, which means one is allowed to query the *exact* value of $f(S)$ for any $S \subseteq N$. However, in many applications, due to the uncertain nature of the objective or the errors involved in the evaluation, one can only access the function value in a distorted or noisy form. For example, Globerson and Roweis [9] pointed out that selecting features to construct a robust learning model is particularly

important in domains with non-stationary feature distributions or input sensor failures. It is known that without any assumption on the noise, direct adaptions of the greedy and local search methods mentioned above may yield arbitrarily poor performance [12]. To address this issue, Hassidim and Singer [11] introduced and studied the following problem of *monotone submodular maximization under noise*: For a monotone submodular function $f$, given a noisy value oracle $\tilde{f}$ satisfying $\tilde{f}(S) = \xi_S f(S)$ for each set $S \subseteq N$, where the noise multiplier $\xi_S$ is independently drawn from a certain distribution, the goal is to find a set $S$ maximizing $f(S)$ under certain constraints. Some applications of this problem are provided in [20] such as revealed preference theory [3] and active learning [6]. Hassidim and Singer [11] showed that under a sufficiently large cardinality constraint, a variant of the greedy algorithm achieves a near-optimal approximation ratio of $1 - \frac{1}{e} - O(\varepsilon)$ for arbitrary $\varepsilon > 0$. The problem becomes more challenging when the cardinality constraint is relatively small since there is less room for mistakes. In a subsequent work, Singer and Hassidim [20] developed another greedy-based algorithm for small cardinality constraints and showed a near-tight approximation guarantee.

Despite these encouraging results, we want to point out two directions along this line of monotone submodular maximization under noise that still have room for improvements:

- In the algorithm from [20], the query complexity to the noisy value oracle is exponential in $\varepsilon^{-1}$, which is costly when a near-optimal solution is needed, i.e., when the parameter $\varepsilon$ is close to 0. Is it possible to obtain near-optimal approximations for monotone submodular maximization under cardinality constraints with the number of queries polynomial in $\varepsilon^{-1}$?

- All previous works in submodular maximization under noise consider only the cardinality constraint, and no approximation guarantee is known under other constraints. [1] Is there any algorithm that can achieve a constant approximation for submodular maximization under noise for more general constraints, such as commonly studied matroid constraints [16, 26]?

In this paper, we provide answers to both questions above.

## 1.1 Our contributions

We study the problem of constrained monotone submodular maximization under noise (Problem 2.6). Following prior works [11, 20], we assume generalized exponential tail noises (Definition 2.5) and consider the solutions subject to cardinality constraints (Definition 2.2) and matroid constraints (Definition 2.3). The main contribution of this work is to show that for optimizing a monotone submodular function under a cardinality constraint, $\left(1 - \frac{1}{e} - O(\varepsilon)\right)$-approximations can be obtained with high probability by querying the noisy oracle only $\text{Poly}\left(n, \frac{1}{\varepsilon}\right)$ times.

**Theorem 1.1 (Informal, see Theorems 4.1 and 4.6).** *Let $\varepsilon > 0$ and assume $n$ is sufficiently large. For any $r \in \Omega\left(\frac{1}{\varepsilon}\right)$, there exists an algorithm that returns a $\left(1 - \frac{1}{e} - O(\varepsilon)\right)$-approximation for the monotone submodular maximization problem under a $r$-cardinality constraint, with probability $1 - o(1)$ and query complexity $\text{Poly}\left(n, \frac{1}{\varepsilon}\right)$ to $\tilde{f}$.*

For a cardinality constraint, this paper and prior works [11, 20] all achieve near-optimal approximations. However, our result is applicable for a larger range $\Omega\left(\frac{1}{\varepsilon}\right)$ of cardinalities than $\Omega\left(\log \log n \cdot \varepsilon^{-2}\right)$ in [11]. Moreover, we only require $\text{Poly}\left(n, \frac{1}{\varepsilon}\right)$ queries to the noisy value oracle, which improves the query complexity $\Omega(n^{\frac{1}{\varepsilon}})$ of [20].

Our main idea to address this problem is to employ a local search procedure, whereas prior methods are all variants of the greedy algorithm. Intuitively, local search is more robust than the greedy since the marginal functions are more sensitive to noise than the value functions. To achieve a sufficient gain in each iteration, the greedy algorithm must identify the element with maximum margin. In contrast, local search only needs to estimate the sets' values. This is why we can improve the query complexity to $\text{Poly}\left(n, \frac{1}{\varepsilon}\right)$ for small cardinality constraints.

We present a unified framework (Algorithm 1) for enhancing the local search to cope with noise. One of the main differences between our framework and the non-oblivious local search proposed by [7] is that we use an approximation of the auxiliary function (Definition 3.2) rather than the exact

---

[1]Note that Singer and Hassidim [20] also provide an algorithm to deal with general matroid constraints. However, we will argue in Section A that their algorithm fails to obtain the approximation guarantee they claim.

one due to the presence of noise. We analyze the impact of the inaccuracies on the approximation performance and query complexity of the local search (Theorem 3.3).

Another difference is the construction of the auxiliary functions used to guide the local search. We construct the auxiliary function not based on the objective function but on smoothing surrogate functions.

A surrogate function $h$ needs to meet two properties: (i) $h$ should depend on an averaging set of size $\text{poly}(n)$, such that $h(S)$ and its noisy analogue $\tilde{h}(S)$ are close for all sets $S$ considered by local search; (ii) $h$ needs to be close to the original function $f$, such that optimizing $h$ can yield a near-optimal solution to optimizing $f$. However, a large averaging set is more likely to induce a large gap between the surrogate and original function, making simultaneous fulfillment of both properties non-trivial.

In this paper we carefully design the smoothing surrogate functions as follows. For a set with size $r \in \Omega\left(\frac{1}{\varepsilon}\right) \cap O\left(n^{1/3}\right)$, we define the smoothing surrogate function $h$ as the expectation of $f$'s value when a random element in $N$ is added to the set (Definition 4.2). This surrogate function is robust for a relatively small cardinality as it is based on a rather large averaging set with size nearly $n$, but too concentrated for a large cardinality close to $n$. Thus, we consider another smoothing surrogate function $h_H$ for size $r \in \Omega\left(n^{1/3}\right)$, defined as the average value combined with all subsets of a certain small-size set $H$ (Definition 4.7). The auxiliary functions constructed on both smoothing surrogates are shown to have almost accurate approximations (Lemma 4.4 and 4.9). Consequently, we can apply our unified local search framework (Algorithm 1) in both cases (Algorithm 2 and 3), and guarantee to achieve nearly tight approximate solutions (Theorem 4.1 and 4.6).

The other contribution of this paper is a constant approximation result for maximizing monotone submodular functions with noisy oracles under general matroid constraints.

**Theorem 1.2 (Informal, see Theorems 5.1 and 5.2).** *Let $\varepsilon > 0$ and assume $n$ is sufficiently large. For any $r \in \Omega\left(\varepsilon^{-1}\log(\varepsilon^{-1})\right)$, there exists an algorithm that returns a $\left(\frac{1}{2}\left(1 - \frac{1}{e}\right) - O(\varepsilon)\right)$-approximation for the monotone submodular maximization problem under a matroid constraint with rank $r$, with probability $1 - o(1)$ and query complexity at most $\text{Poly}\left(n, \frac{1}{\varepsilon}\right)$ to $\tilde{f}$.*

To the best of our knowledge, this is the first result showing that constant approximation guarantees are obtainable under general matroid constraints in the presence of noise. To cope with noise, one common approach for cardinality constraints is to incorporate some extra elements to gain robustness and include these elements in the final solutions. However, for a matroid, additional elements may undermine the independence of a set. To address this difficulty, we develop a technique for comparing the values of independent sets in the presence of noise, which allows us to select either the local search solutions or the additional elements for robustness and leads to an approximation ratio of $\frac{1}{2}\left(1 - \frac{1}{e}\right)$.

## 1.2 Related work

Research has been conducted on monotone submodular maximization in the presence of noise. We say a noisy oracle is inconsistent if it returns different answers when repeatedly queried. For inconsistent oracles, noise often does not present a barrier to optimization, since concentration assumptions can eliminate the noise after a sufficient number of queries [21, 13]. When identical queries always obtain the same answer, the problem becomes more challenging. Aside from the i.i.d noise adopted in [11, 20] and this paper, Horel and Singer [12] study submodular optimization under noise adversarially generated from $[1 - \varepsilon/r, 1 + \varepsilon/r]$, where the greedy algorithm achieves a ratio of $1 - 1/e - O(\varepsilon)$. No algorithm can obtain a constant approximation if noise is not bounded in this range.

## 2 The model

This section formally defines our model (Problem 2.6) of maximizing a monotone submodular function (Definition 2.1) under a cardinality constraint (Definition 2.2) or a matroid constraint (Definition 2.3), given access to a noisy value oracle (Definition 2.4). Let $N$ be the ground set with size $|N| = n$, and we use the shorthands $S + x = S \cup \{x\}$ and $S - x = S \backslash \{x\}$ throughout this paper. We first review the definition of monotone submodular functions.

**Definition 2.1 (Monontone submodular functions).** A function $f : 2^N \to \mathbb{R}_{\geq 0}$ is monotone submodular if 1) (monotonicity) $f(A) \leq f(B)$ for any $A \subseteq B \subseteq N$; 2) (submodularity) for any subset $A, B \subseteq N$ and $x \in N$: $f(A + x) - f(A) \geq f(B + x) - f(B)$.

There are various examples of monotone submodular functions in optimization, such as budget additive functions, coverage functions, cut functions and rank functions [1]. Since the description of a submodular function may be exponential in the size $N$, we usually assume access of a *value oracle* that answers $f(S)$ for each $S \subseteq N$. Let $\mathcal{I}$ denote a collection of feasible subsets $S \subseteq N$. The goal of a constrained monotone submodular maximization is to find a subset $S \subseteq \mathcal{I}$ to maximize $f(S)$. The objective $f$ is further assumed to be normalized, i.e., $f(\varnothing) = 0$. We consider two types of constraints in our models: cardinality constraints and matroid constraints.

**Definition 2.2 (Cardinality constraints).** Given a ground set $N$ and an integer $r \geq 1$, a cardinality constraint is of the form $\mathcal{I}(r) = \{S \subseteq N : |S| \leq r\}$.

**Definition 2.3 (Matroids and Matroid constraints).** Given a ground set $N$, a matroid $\mathcal{M}$ is represented by an ordered pair $(N, \mathcal{I}(\mathcal{M}))$ satisfying that 1) $\varnothing \in \mathcal{I}(\mathcal{M})$; 2) If $I \in \mathcal{I}(\mathcal{M})$ and $I' \subseteq I$, then $I' \in \mathcal{I}(\mathcal{M})$; 3) If $I_1, I_2 \subseteq \mathcal{I}(\mathcal{M})$ and $|I_1| < |I_2|$, then there must exist an element $e \in I_2 \setminus I_1$ such that $I_1 \cup \{e\} \in \mathcal{I}(\mathcal{M})$. Each $I \in \mathcal{I}(\mathcal{M})$ is called an independent set. The maximum size of an independent set is called the rank of $\mathcal{M}$. We call the collection $\mathcal{I}(\mathcal{M})$ a matroid constraint.

Assume we are given a *membership oracle* of $\mathcal{I}(\mathcal{M})$ that for any set $S \subseteq N$ answers whether $S \in \mathcal{I}(\mathcal{M})$. As a widely used combinatorial structure, there is an extensive study on matroids [28, 19, 27]. Common matroids include uniform matroids, partition matroids, regular matroids, etc. See [19] for more discussions. Specifically, a uniform matroid constraint is equivalent to a cardinality constraint, implying that cardinality constraints are a special case of matroid constraints.

**Noisy value oracle.** It is well known that given an exact value oracle to $f$, for any $\varepsilon > 0$, there exists a randomized $(1 - 1/e - \varepsilon)$-approximate algorithm for the submodular optimization problem under a matroid constraint, i.e., $\max_{S \subseteq \mathcal{I}} f(S)$ [2, 7]. However, as discussed earlier, the value oracle of $f$ may be imperfect, and we may only have a noisy value oracle $\tilde{f}$ instead of $f$. We consider the following noisy value oracle that has also been investigated in [12, 11].

**Definition 2.4 ((Multiplicative) noisy value oracle [11]).** We call $\tilde{f} : 2^N \to \mathbb{R}_{\geq 0}$ a (multiplicative) noisy value oracle of $f$, if there exists some distribution $\mathcal{D}$ s.t. for any $S \subseteq N$, $\tilde{f}(S) = \xi_S f(S)$ where $\xi_S$ is i.i.d. drawn from $\mathcal{D}$.

Throughout this paper, we consider a general class of noise distributions, called generalized exponential tail distributions, defined in [20].

**Definition 2.5 (Generalized exponential tail distributions [20]).** A noise distribution $\mathcal{D}$ has a *generalized exponential tail* if there exists some $x_0$ such that for every $x > x_0$ the probability density function $\rho(x) = e^{-g(x)}$, where $g(x) = \sum_i c_i x^{\gamma_i}$ for some (not necessarily integers) $\gamma_0 \geq \gamma_1 \geq \ldots, s.t. \gamma_0 \geq 1$ and $c_0 > 0$. If $\mathcal{D}$ has bounded support we only require that either it has an atom at its supremum or that $\rho$ is continuous and non-zero at the supremum.

W.l.o.g., we assume $\mathbb{E}[\mathcal{D}] = 1$.[2] As mentioned in [20], the class of generalized exponential tail distributions contains Gaussian distributions, exponential distributions, and all distributions with bounded support that is independent of $n$.

**The model.** We are ready to propose the main problem, which has already been considered in [11, 20].

**Problem 2.6 (Constrained submodular optimization under noise).** *Given a noisy value oracle* $\tilde{f} : 2^N \to \mathbb{R}_{\geq 0}$ *(with a certain generalized exponential tail distribution $\mathcal{D}$) to an underlying monotone submodular function $f$, and a cardinality constraint $\mathcal{I} = \mathcal{I}(r)$ or a matroid constraint $\mathcal{I} = \mathcal{I}(\mathcal{M})$, the goal is to find $S \subseteq \mathcal{I}$ to maximize $f(S)$.*

## 3 Local search with approximate evaluation oracles to auxiliary functions

This section proposes a non-oblivious local search framework with a noisy value oracle (Algorithm 1), and gives an analysis (Theorem 3.3) of its performance. This algorithm is a generalization of that

---

[2]Otherwise, we can scale $\mathcal{D}$ to be $\mathcal{D}' = \mathcal{D}/\mathbb{E}[\mathcal{D}] = 1$.

in [7] and will be used as a meta-algorithm for solving Problem 2.6 in Sections 4 and 5. Roughly speaking, a local search framework first constructs a "good" initial solution via a standard greedy algorithm and then iteratively improves the solution w.r.t. to an auxiliary function (Definition 3.1) by swapping an element at each step. The following auxiliary function proposed by [7] is a linear combination of the original function over all subsets.

**Definition 3.1 (Auxiliary function [7]).** Given a monotone submodular function $h : 2^N \to \mathbb{R}_{\geq 0}$, its auxiliary function $\varphi_h : 2^N \to \mathbb{R}_{\geq 0}$ is defined as $\varphi_h(S) = \sum_{T \subseteq S} m_{|S|-1, |T|-1} \cdot h(T)$, where $m_{s,t} = \int_0^1 e^p (e-1)^{-1} p^t (1-p)^{s-t} dp$ for all $s \geq t \geq 0$.

We use $h$ instead of $f$ because we may do local search on certain smoothing surrogate function $h$ of $f$ (see examples in Definitions 4.2 and 4.7). Note that $\varphi_h$ is also a monotone submodular function. Given an exact value oracle of $\varphi_h$ and a matroid constraint $\mathcal{I}(\mathcal{M})$, Filmus and Ward [7] show that there exists a local search algorithm that outputs a $(1-1/e)$-approximate solution for the optimization problem $\max_{S \subseteq \mathcal{I}(\mathcal{M})} h(S)$. However, as mentioned before, we do not have an exact value oracle of $h$. Thus, it may not be possible to construct an exact value oracle of $\varphi_h$. To understand how these inaccuracies can affect the performance of the local search algorithm, we introduce the following approximation for the auxiliary function $\varphi_h$.

**Definition 3.2 ($(\alpha, \delta, \mathcal{I}(\mathcal{M}))$-approximation of $\varphi_h$).** Given an auxiliary function $\varphi_h$, a matroid constraint $\mathcal{I}(\mathcal{M})$ and constants $\alpha, \delta > 0$, we say a randomized function $\widehat{\varphi_h}$ is an $(\alpha, \delta, \mathcal{I}(\mathcal{M}))$-approximation of $\varphi_h$ if for any independent set $A \in \mathcal{I}(\mathcal{M})$,

$$\mathbb{P}\left[ |\widehat{\varphi_h}(A) - \varphi_h(A)| > \alpha \cdot \max_{S \in \mathcal{I}(\mathcal{M})} \varphi_h(S) \right] \leq \delta.$$

Intuitively, $\widehat{\varphi_h}$ is a randomized version of $\varphi_h$ with an additive concentration guarantee, where the randomness may come from noise of $h$ or sampling error for estimating $\varphi_h$. As $\alpha, \delta$ tend to 0, $\widehat{\varphi_h}$ is a better approximation of $\varphi_h$. Specifically, when $\alpha = \delta = 0$ and $\mathcal{I} = 2^N$, $\widehat{\varphi_h}$ is exactly equivalent to $\varphi_h$. Now we are ready to provide our local search framework (Algorithm 1), which is a generalization of [7, Algorithm 2]. The main difference is that we use an approximation $\widehat{\varphi_h}$ instead of $\varphi_h$ for evaluation at each greedy step (Line 4) and local search step (Line 12). The performance of Algorithm 1 is summarized in Theorem 3.3.

---

**Algorithm 1:** Noisy local search ($\texttt{NLS}(\widehat{\varphi_h}, \mathcal{I}(\mathcal{M}), \Delta)$)

**Input :** A matroid constraint $\mathcal{I}(\mathcal{M})$ of rank $r \geq 1$, a value oracle to an $(\alpha, \delta, \mathcal{I}(\mathcal{M}))$-approximation $\widehat{\varphi_h}$ of $\varphi_h$, a stepsize $\Delta \in (0, 1/2)$.

1   Set $I \leftarrow \left\lceil \log_{1+\Delta} \left( \frac{2(1+\alpha)}{1-2(r+1)\alpha} \right) \right\rceil$
2   Initialize: $U_0 \leftarrow \varnothing$, $i = 1$
3   **while** $i \leq r$ **do**
4      $u_i \leftarrow \arg\max_{e: U_{i-1}+e \in \mathcal{I}(\mathcal{M})} \widehat{\varphi_h}(U_{i-1} + e)$
5      $U_i \leftarrow U_{i-1} + u_i$
6      $i \leftarrow i + 1$
7   $S_0 \leftarrow U_r$                         ▷ Initial solution by greedy
8   **for** $i = 0$ *to* $I - 1$ **do**            ▷ Local search for $I$ iterations
9      **for** *each element $x \in S_i$ and $y \in N \setminus S_i$* **do**
10         $S_i' \leftarrow S_i - x + y$
11         **if** $S_i' \in \mathcal{I}(\mathcal{M})$ **then**
12            **if** $\widehat{\varphi_h}(S_i') \geq (1+\Delta) \cdot \widehat{\varphi_h}(S_i)$ **then**    ▷ An improved solution $S_i'$ was found
13               $S_{i+1} \leftarrow S_i'$
14               break and continue to the next iteration of $i$
15   **return** $S_I$

---

**Theorem 3.3 (Performance of noisy local search).** *Let $I = \log_{1+\Delta} \left( \frac{2(1+\alpha)}{1-2(r+1)\alpha} \right)$. With probability at least $1 - (I+1)rn\delta$, the output $S_I$ of Algorithm 1 is a $\left(1 - \frac{1}{e}\right)\left(1 - r\ln(er)\left((2+\Delta)\alpha + \Delta\right)\right)$-approximate solution for problem $\max_{S \subseteq \mathcal{I}(\mathcal{M})} h(S)$, with at most $(I+1)rn$ calls to $\widehat{\varphi_h}$.*

We defer the proof to Section C. Roughly, given a nearly accurate approximation $\widehat{\varphi_h}$, we can show that the initial solution $S_0$ is likely to be a constant approximation for $\max_{S \in \mathcal{I}(\mathcal{M})} \varphi_h(S)$ (Lemma C.1). This guarantees that we will reach a local optimal solution w.r.t. $\varphi_h$ in $I$ iterations with high probability (Claim C.3). Then combining with basic properties of $\varphi_h$ shown in [7] and the fact that $\widehat{\varphi_h}$ is "close" to $\varphi_h$, we obtain Theorem 3.3.

**Corollary 3.4.** *Assume* $n \in \Omega\left(\exp\left(\left(\frac{1}{\varepsilon}\right)^{O(1)}\right)\right)$, *if* $\alpha, \Delta = \frac{\varepsilon}{4r \log r}$, *we have* $I \leq \frac{5r \ln r}{\varepsilon}$, *and with probability at least* $1 - (I+1)rn\delta$, *the output* $S_I$ *of Algorithm 1 is a* $\left(1 - \frac{1}{e} - \varepsilon\right)$*-approximate solution for problem* $\max_{S \subseteq \mathcal{I}(\mathcal{M})} h(S)$, *with at most* $(I+1)\,rn$ *calls to* $\widehat{\varphi_h}$.

# 4 Our algorithms and main theorems for cardinality constraints

For different ranges of cardinalities, this section presents algorithms (Algorithm 2 and 3) that return a $\left(1 - \frac{1}{e} - O(\varepsilon)\right)$-approximation for Problem 2.6 with only $\text{Poly}\left(n, \frac{1}{\varepsilon}\right)$ queries to the noisy oracle $\tilde{f}$. The analyses (Theorem 4.1 and 4.6) of the algorithms constitute a proof of Theorem 1.1.

## 4.1 Algorithmic results for cardinality constraints when $r \in \Omega\left(\frac{1}{\varepsilon}\right) \cap O\left(n^{\frac{1}{3}}\right)$

We first present an algorithm (Algorithm 2) and its analysis (Theorem 4.1) that are applicable for all $r \in \Omega\left(\frac{1}{\varepsilon}\right) \cap O\left(n^{\frac{1}{3}}\right)$ when $n$ is sufficiently large. All missing proofs can be found in Section D.

**Theorem 4.1 (Algorithmic results for cardinality constraints when $r \in \Omega\left(\frac{1}{\varepsilon}\right) \cap O\left(n^{\frac{1}{3}}\right)$).** *Let* $\varepsilon > 0$ *and assume* $n \in \Omega\left(\exp\left(\left(\frac{1}{\varepsilon}\right)^{O(1)}\right)\right)$ *is sufficiently large. For any* $r \in \Omega(\frac{1}{\varepsilon}) \cap O\left(n^{\frac{1}{3}}\right)$, *there exists an algorithm that returns a* $\left(1 - \frac{1}{e} - O(\varepsilon)\right)$*-approximation for Problem 2.6 under a* $r$*-cardinality constraint, with probability at least* $1 - O\left(\frac{1}{\log n}\right)$ *and query complexity at most* $O\left(r^2 \log^2 r \cdot n^{\frac{3}{2}} \varepsilon^{-1} \max\{r, \log n\}\right)$ *to* $\tilde{f}$.

The assumption that $n$ is sufficiently large is necessary and has also been adopted in prior works on noisy submodular optimization [11, 20]. We achieve a tight approximation ratio of $1 - 1/e - O(\varepsilon)$. Furthermore, we only require $\text{Poly}\left(n, 1/\varepsilon\right)$ queries to $\tilde{f}$, in contrast to $\Omega\left(n^{1/\varepsilon}\right)$ for the prior greedy algorithm [20].

### 4.1.1 Useful notations and useful facts for Theorem 4.1

Our algorithm and analysis are based on the following smoothing surrogate function $h$.

**Definition 4.2 (Smoothing surrogate function I).** For any set $S \subseteq N$, we define the smoothing surrogate function $h(S)$ as the expectation of $f(S + e)$ over a random element $e \in N$, i.e., $h(S) = \frac{1}{n} \sum_{e \in N} f(S + e)$.

The surrogate function $h(S)$ is robust to noise when $|S|$ is relatively small since at this time $h(S)$ is based on a rather large averaging set with size nearly $n$. Note that $h(S)$ becomes too concentrated on $f(S)$ as $|S| \approx n$, and hence, we consider this surrogate $h$ for the range $r \in O\left(n^{\frac{1}{3}}\right)$. We now show that $h$ shares some basic properties with $f$.

**Lemma 4.3 (Properties of $h$).** *The smoothing surrogate function $h$ is monotone and submodular.*

Note that $h$ is implicitly constructed since we only have a value oracle to $\tilde{f}$ instead of $f$. Hence, we construct an approximation of its auxiliary function $\varphi_h$; summarized by the following lemma.

**Lemma 4.4 (Approximation of $\varphi_h$).** *Let* $\alpha, \delta \in (0, 1/2)$ *and assume* $n \in \Omega(\alpha^{-2} \log(\delta^{-1}))$. *There exists a value oracle $\mathcal{O}$ to an* $(\alpha, \delta, \mathcal{I}(r-1))$*-approximation* $\widehat{\varphi_h}$ *of* $\varphi_h$, *which answering* $\mathcal{O}(A)$ *queries at most* $M = \Theta\left(\log r \cdot n^{\frac{1}{2}} \max\{r, \log n\}\right)$ *times to* $\tilde{f}$ *for each set* $A \in \mathcal{I}(r-1)$.

The above lemma indicates that for a sufficiently large $n$, we can achieve an arbitrary accurate approximation $\widehat{\varphi_h}$ of $\varphi_h$, which queries $\tilde{f}$ only $\tilde{O}(rn^{\frac{1}{2}})$ times for each $\widehat{\varphi_h}(A)$.

More precisely, the relationship among $\alpha$, $\delta$ and $n$ can be expressed as $\alpha^{-2} \ln(4\delta^{-1}) \le c \cdot n\kappa^{-2}$ (See Lemma D.5), where $\kappa$ is the sub-exponential norm (Definition D.2) of the noise distribution $\mathcal{D}$ depending on the parameters $c_i, \gamma_i$ of density function, and $c$ is an absolute constant. A small $\kappa$ indicates a concentrated noise distribution. Generally speaking, the more concentrated the noise is ($\kappa \to 0$), the more accurate approximation we are able to obtain ($\alpha, \delta \to 0$).

### 4.1.2 Algorithm for Theorem 4.1

---

**Algorithm 2:** Noisy local search under a small cardinality constraint

---

**Input :** a value oracle to $\tilde{f}$, budget $r \in \Omega\left(\frac{1}{\varepsilon}\right) \cap O\left(n^{\frac{1}{3}}\right)$ and $\varepsilon \in (0, 1/2)$

**1** Let $\widehat{\varphi_h}$ be a $\left(\alpha = \frac{\varepsilon}{4r \ln r}, \delta = \frac{1}{(I+1)(r-1)n^2}, \mathcal{I}(r-1)\right)$-approximation of $\varphi_h$ as in Lemma 4.4

**2** $S_L \leftarrow \text{NLS}\left(\widehat{\varphi_h}, \mathcal{I}(r-1), \Delta = \frac{\varepsilon}{4r \ln r}\right)$        ▷ `Local search phase`

**3** $S_M \leftarrow S_L + \arg\max_{e \in N \setminus S_L} \tilde{f}(S_L + e)$        ▷ $\tilde{f}$-`maximization phase`

**4 return** $S_M$

---

We present Algorithm 2 that contains two phases: a local search phase (Line 2) and a $\tilde{f}$-maximization phase (Line 3). We first run a non-oblivious local search procedure (Algorithm 1) under the $(r-1)$-cardinality constraint (Line 2). We set $\delta = \frac{1}{(I+1)(r-1)n^2}$ by Corollary 3.4 when applying Algorithm 1 to obtain a $\left(1 - \frac{1}{e} - \varepsilon\right)$-approximation solution $S_L$ for $h$-maximization. At the $\tilde{f}$-maximization phase (Line 3), we obtain $S_M$ by selecting an additional element $e \in N \setminus S_L$ that maximizes $\tilde{f}(S_L + e)$. This guarantees that $f(S_M) \ge (1 - \varepsilon)h(S_L)$ with high probability (Lemma 4.5), which results in the approximation ratio $1 - \frac{1}{e} - O(\varepsilon)$ in Theorem 4.1.

Given the smoothing surrogate function $h(S)$ (Definition 4.2), a natural idea is to simply apply local search (Algorithm 1) to optimize it under the $r$-cardinality constraint. However, this idea does not yield a provable approximation of $\max_{S:|S| \le r} f(S)$, since it is not easy to control the contribution of the additional element $e \in N$ to $h(S)$ and it is possible that $h(S) \gg f(S)$. To handle this difficulty, Algorithm 2 first runs a local search procedure under the $(r-1)$-cardinality constraint, and then selects an additional element with a "large enough" margin at the $\tilde{f}$-maximization phase. This idea enables us to control the loss induced by the surrogate $h$ within $\frac{1}{r} \cdot \max_{S:|S| \le r} f(S)$.

### 4.1.3 Proof sketch of Theorem 4.1

We first analyze the query complexity and then prove the approximation performance.

**Query complexity of Algorithm 2.** In Line 3, we make $(n - r + 1)$ calls to the noisy oracle $\tilde{f}$ in total. By Corollary 3.4, the total number of queries to $\widehat{\varphi_h}$ in Line 2 is at most $(I+1)(r-1)n$ and $I \le \frac{5r \ln r}{\varepsilon}$. Combining with Lemma 4.4, the query complexity to $\tilde{f}$ is upper bounded by $(n-r+1) + M(I+1)(r-1)n = O\left(r^2 \log^2 r \cdot n^{\frac{3}{2}} \varepsilon^{-1} \max\{r, \log n\}\right)$. This matches Theorem 4.1.

**Approximation ratio and success probability of Algorithm 2.** Let $O_h \in \arg\max_{S \in \mathcal{I}(r-1)} h(S)$ represent the $(r-1)$-set whose value of $h$ is the largest. Following from Theorem 3.3, we have $h(S_L) \ge \left(1 - \frac{1}{e} - \varepsilon\right) h(O_h)$ with probability at least $1 - \frac{1}{n}$. We denote by $O_f \in \arg\max_{S \in \mathcal{I}(r)} f(S)$ the optimal solution to $f$. By the submodularity of $f$, $O_f$ has a subset $\tilde{O}_f$ with $r - 1$ elements such that $f(\tilde{O}_f) \ge \left(1 - \frac{1}{r}\right) f(O_f)$. Then we have

$$h(S_L) \ge \left(1 - \frac{1}{e} - \varepsilon\right) h(\tilde{O}_f) \ge \left(1 - \frac{1}{e} - \varepsilon\right) f(\tilde{O}_f) \ge \left(1 - \frac{1}{r}\right)\left(1 - \frac{1}{e} - \varepsilon\right) f(O_f), \quad (1)$$

where the first inequality follows from $h(S_L) \ge \left(1 - \frac{1}{e} - \varepsilon\right) h(O_h)$ and the second from monotonicity of $f$. Since $r$ is assumed to be $\Omega\left(\frac{1}{\varepsilon}\right)$, $h(S_L)$ is a $\left(1 - \frac{1}{e} - O(\varepsilon)\right)$-approximation of $f(O_f)$.

Recall that $h(S_L)$ is the expectation of $f(S_L + e)$ over a random element $e \in N$. Ineq. (1) already proves a claim that uniformly randomly adding an element $e \in N$ to $S_L$ achieves an approximation ratio of $1 - \frac{1}{e} - O(\varepsilon)$ in expectation for maximizing $f$, i.e., $\mathbb{E}[f(S_L + e)] \ge \left(1 - \frac{1}{e} - O(\varepsilon)\right) f(O_f)$. Moreover, we can convert this claim to a with-high-probability claim by the following lemma.

**Lemma 4.5 (Approximation at the $\tilde{f}$-maximization phase).** *With probability $1 - O\left(\frac{1}{\log n}\right)$, we have $f(S_M) \geq (1 - \varepsilon)h(S_L)$.*

This lemma indicates that a *bad* element $e \in N \setminus S_L$ with $f(S_L + e) < (1 - \varepsilon)h(S_L)$ is unlikely to be chosen at the $\tilde{f}$-maximization phase. This is because the local search procedure guarantees that the number of *good* elements with $f(S_L + e) \geq (1 - \varepsilon)h(S_L)$ is almost of the same order as the bad ones. Consequently, we can show that the selected element is likely to be good due to the generalized exponential tail noise, using similar proof idea as that of [20, Lemma 3.4]. By Ineq. (1) and Lemma 4.5, we complete the proof.

## 4.2 Algorithmic results for cardinality constraints when $r \in \Omega\left(n^{\frac{1}{3}}\right)$

We present in this subsection an algorithm (Algorithm 3) and its analysis (Theorem 4.6) that can be applied to $r \in \Omega\left(n^{\frac{1}{3}}\right)$ when $n$ is sufficiently large. All missing proofs can be found in Section E.

**Theorem 4.6 (Algorithmic results for cardinality constraints when $r \in \Omega\left(n^{\frac{1}{3}}\right)$).** *Let $\varepsilon > 0$ and assume $n \in \Omega\left(\frac{1}{\varepsilon^4}\right)$ is sufficiently large. For any $r \in \Omega\left(n^{\frac{1}{3}}\right)$, there exists an algorithm that returns a $\left(1 - \frac{1}{e} - O(\varepsilon)\right)$-approximation for Problem 2.6 under a $r$-cardinality constraint, with probability $1 - O\left(\frac{1}{n^2}\right)$ and query complexity at most $O(n^6\varepsilon^{-1})$ to $\tilde{f}$.*

The above theorem addresses the remaining range of $r$ in Theorem 4.1, and also achieves a tight approximation ratio of $1 - \frac{1}{e} - O(\varepsilon)$ with $\mathrm{Poly}(n, 1/\varepsilon)$ queries to $\tilde{f}$.

### 4.2.1 Useful notations and useful facts for Theorem 4.6

Our algorithm and analysis in this subsection are based on another smoothing surrogate function $h$.

**Definition 4.7 (Smoothing surrogate function II).** Given a subset $H \subseteq N$, for any set $S \subseteq N \setminus H$, we define the smoothing surrogate function $h_H(S)$ as $h_H(S) = \sum_{H_j \subseteq H} f(S \cup H_j)/2^{|H|}$.

Intuitively, if $|H|$ is sufficiently large, the surrogate function $h$ is robust to noise as it is based on a large averaging set with size $2^{|H|}$. Throughout the rest of this subsection, we consider the case that $|H| \in \Omega(\log n)$. Similar to Section 4.1.1, we give some basic properties of $h_H$.

**Lemma 4.8 (Properties of $h_H$).** *For any $H \subseteq N$, the smoothing surrogate function $h_H$ is monotone and submodular, and for all $S \subseteq N \setminus H$, $h_H(S) \geq \frac{1}{2}f(S \cup H) + \frac{1}{2}f(S)$.*

Given a cardinality $r$ and a set $H \subseteq N$, we define a $(r - |H|)$-cardinality constraint confined on $N \setminus H$ as $\mathcal{I}_H(r) = \{S \subseteq N \setminus H : |S| = r - |H|\}$. Similar to Section 4.1.1, we provide the following lemma indicating that the auxiliary function $\varphi_{h_H}$ can be well approximated.

**Lemma 4.9 (Approximation of $\varphi_{h_H}$).** *Let $\varepsilon > 0$ and assume $n \in \Omega\left(\varepsilon^{-4}\right)$. For a set $H$ with $|H| \geq 3\ln n$, there exists a value oracle $\mathcal{O}$ to a $\left(\alpha = \frac{\varepsilon}{4r\ln r}, \delta = \frac{3}{n^6}, \mathcal{I}_H(r)\right)$-approximation $\widehat{\varphi_{h_H}}$ of $\varphi_{h_H}$, which answering $\mathcal{O}(A)$ queries $\tilde{f}$'s oracle $O\left(r\varepsilon^{-1}\log^{\frac{5}{2}} n\log^2 r\right)$ times for each set $A \in \mathcal{I}_H(r)$.*

The above lemma indicates that for sufficiently large $n$ and $|H|$, we can achieve an almost accurate approximation $\widehat{\varphi_{h_H}}$ of $\varphi_{h_H}$, which queries $\tilde{f}$ only $\widetilde{O}(n\varepsilon^{-1})$ times for each $\widehat{\varphi_{h_H}}(A)$. By Corollay 3.4, we set $\delta = \frac{3}{n^6}$ when applying Algorithm 1.

### 4.2.2 Algorithm for Theorem 4.6

We present Algorithm 3 that mainly contains a local search phase (Line 3). The main difference from Algorithm 2 is that Algorithm 3 arbitrarily select a redundant set $H$ (Line 1), and use this set to construct a smoothing surrogate function $h_H$ for local search (Line 3). Then Algorithm 3 returns $S_L \cup H$ directly (Line 4) without a $\tilde{f}$-maximization phase.

---
**Algorithm 3:** Noisy local search under large cardinality constraint
---
**Input :** a value oracle to $\tilde{f}$, budget $r \in \Omega\left(n^{\frac{1}{3}}\right)$, and $\varepsilon \in (0, 1/2)$
1   Arbitrarily select a subset $H \subseteq N$ with $|H| = \lceil 3 \ln n \rceil$
2   Let $\widehat{\varphi_{h_H}}$ be a $\left(\alpha = \frac{\varepsilon}{4r \ln r}, \delta = \frac{3}{n^6}, \mathcal{I}_H(r)\right)$-approximation of $\varphi_{h_H}$ as in Lemma 4.9
3   $S_L \leftarrow \text{NLS}\left(\widehat{\varphi_{h_H}}, \mathcal{I}_H(r), \Delta = \frac{\varepsilon}{4r \ln r}\right)$             ▷ Local search phase
4   **return** $S_L \cup H$
---

### 4.2.3   Proof sketch of Theorem 4.6

Again, we first analyze the query complexity and then prove the approximation performance.

**Query complexity of Algorithm 3.** Since $n \in \Omega\left(\frac{1}{\varepsilon^4}\right)$, we have $I \in O(n^2)$ in Corollary 3.4. Hence Line 4 calls the oracle of $\widehat{\varphi_{h_H}}$ at most $O(rn^3)$ times by Corollary 3.4, and each call to $\widehat{\varphi_{h_H}}$ queries $\tilde{f}$ at most $O\left(r\varepsilon^{-1} \log^{\frac{5}{2}} n \log^2 r\right)$ times by Lemma 4.9. Thus we query $\tilde{f}$ at most $O(n^6 \varepsilon^{-1})$ times.

**Approximation ratio and success probability of Algorithm 3.** Corollary 3.4, we have $h_H(S_L) \geq \left(1 - \frac{1}{e} - \varepsilon\right) \cdot \max_{S \subseteq \mathcal{I}_H(r)} h_H(S)$ with probability $1 - O\left(\frac{1}{n^2}\right)$. Furthermore, $f(S_L \cup H) \geq h_H(S_L)$ follows from the monotonicity of $f$. Then we can demonstrate the approximation performance of Algorithm 3 by the following lemma.

**Lemma 4.10.** *Let* $O^\star = \arg\max_{S \in \mathcal{I}(r)} f(S)$. *We have* $\max_{S \in \mathcal{I}_H(r)} h_H(S) \geq \left(1 - \frac{|H|}{r}\right) f(O^\star)$.

Roughly speaking, there exists a subset $A \subseteq O^\star$ of size $|H|$ such that $f(O^\star \backslash A) \geq \left(1 - \frac{|H|}{r}\right) f(O^\star)$ by submodularity of $f$. The claim follows from $\max_{S_L \in \mathcal{I}_H(r)} h_H(S) \geq h(O^\star \backslash A) \geq f(O^\star \backslash A)$. Combining the fact that $f(S_L \cup H) \geq \left(1 - \frac{1}{e} - \varepsilon\right) \max_{S \subseteq \mathcal{I}_H(r)} h_H(S)$ with Lemma 4.10, we complete the proof of Theorem 4.6.

## 5   Our algorithms and main theorems for general matroid constraints

In this section, we consider Problem 2.6 under matroid constraints. Similarly, for different ranges of matroid ranks, we prove there are $(1 - \frac{1}{e} - O(\varepsilon))/2$-approximate algorithms (Theorems 5.1 and 5.2). W.l.o.g., we assume that all single elements are feasible for the given matroid constraints.

**Theorem 5.1** (Algorithmic results for matroid constraints with rank $r \in \Omega\left(\frac{1}{\varepsilon} \log\left(\frac{1}{\varepsilon}\right)\right) \cap O\left(n^{\frac{1}{3}}\right)$)**.**

*Let $\varepsilon > 0$ is sufficiently small and assume $n \in \Omega\left(\exp\left(\left(\frac{1}{\varepsilon}\right)^{O(1)}\right)\right)$ is sufficiently large. For any $r \in \Omega(\varepsilon^{-1} \log(\varepsilon^{-1})) \cap O\left(n^{\frac{1}{3}}\right)$, there exists an algorithm that returns a $\left(\left(1 - \frac{1}{e}\right)/2 - O(\varepsilon)\right)$-approximation for Problem 2.6 under a matroid constraint $\mathcal{I}(\mathcal{M})$ with rank $r$, with probability at least $1 - O\left(\varepsilon^4\right)$ and query complexity at most $O\left(r^2 \log^2 r \cdot n^{\frac{3}{2}} \varepsilon^{-1} \max\{r, \log n\}\right)$ to $\tilde{f}$.*

The proof can be found in Section F. The main difference from Theorem 4.1 is that the approximation ratio is $\left(1 - \frac{1}{e}\right)/2 - O(\varepsilon)$ instead of $1 - \frac{1}{e} - O(\varepsilon)$. The reason is that to maintain the feasibility of the output, we may not be able to add an element to $S_L$ as in Line 3 of Algorithm 2. To address this issue, we design a comparison procedure (Lines 4-8 of Algorithm 7) to evaluate the value of $f(S_L)$ and $f(S_M \setminus S_L)$ and output the larger one, which results in a loss on the approximation ratio. Moreover, the failure probability of the comparison procedure is upper bounded by $O(\varepsilon^4)$.

**Theorem 5.2** (Algorithmic results for matroid constraints with rank $r \in \Omega\left(n^{\frac{1}{3}}\right)$)**.** *Let $\varepsilon > 0$ and assume $n \in \Omega\left(\frac{1}{\varepsilon^4}\right)$ is sufficiently large. For $r \in \Omega\left(n^{\frac{1}{3}}\right)$, there exists an algorithm returning a $\left(\left(1 - \frac{1}{e}\right)/2 - O(\varepsilon)\right)$-approximation for Problem 2.6 under a matroid constraint $\mathcal{I}(\mathcal{M})$ with rank $r$, with probability $1 - O\left(\frac{1}{n^2}\right)$ and query complexity $O(n^6 \varepsilon^{-1})$ to $\tilde{f}$. Specifically, for a strongly base-orderable matroid (Definition H.1), the approximation ratio can be improved to $1 - \frac{1}{e} - O(\varepsilon)$.*

The proof for general matroid constraints can be found in Section G. The main difference from Theorem 4.6 is again the approximation ratio. For maintaing the feasibility, we divide an arbitrary base into two equal-size subsets, apply Algorithm 3 to both subsets, and output the larger one. Consequently, we achieve a $\left(\left(1 - \frac{1}{e}\right)/2 - O(\varepsilon)\right)$-approximation ratio. Specifically, for a strongly base-orderable matroid, we can use its good exchangable property to maintain the feasibility without sacrificing the approximation ratio; see Section H for a complete proof.

## 6  Discussions

For any cardinality $r \geq 2$, Algorithms 2 and 3 return a $\left(\left(1 - \frac{1}{r}\right)\left(1 - \frac{1}{e}\right) - O(\varepsilon)\right)$-approximation for monotone submodular maximization problem under noise with query complexity $\text{Poly}\left(n, \frac{1}{\varepsilon}\right)$. When $r \in \Omega\left(\frac{1}{\varepsilon}\right)$, the approximation ratio can be written as $\left(1 - \frac{1}{e}\right) - O(\varepsilon)$ (Theorem 1.1). However, if $r \in O\left(\frac{1}{\varepsilon}\right)$, then this ratio is not close to the optimal one. In this case, Singer and Hassidim [20] provide an algorithm that achieves $\left(1 - \frac{1}{r}\right)$-approximation with query complexity $\Omega(n^r)$. It remains open whether there is an algorithm that returns $\left(1 - \frac{1}{e} - O(\varepsilon)\right)$-approximation with query complexity $\text{Poly}\left(n, \frac{1}{\varepsilon}\right)$ for $r \in O\left(\frac{1}{\varepsilon}\right)$. The main challenges of our approach to handle this range of $r$ is that the introduction of surrogates (Definition 4.2) inevitably results in a loss of $\frac{1}{r}$ in approximation ratio. For the special case of $r = 1$, one of the few known results is an algorithm by [20] achieving a $\frac{1}{2}$-approximation guarantee in expectation, which is information theoretically tight. To the best of our knowledge, there is no with-high-probability result for this case.

Another limitation of this work is that we only consider independent noise. For non-i.i.d. noise, Hassidim and Singer [11] indicate that no algorithm can achieve a constant approximation when the noise multipliers are arbitrarily correlated across sets. Considering this, it may be worthwhile to consider special cases of correlated distributions for which optimal guarantees can be obtained. One of them is a model called $d$-correlated noise [11]: a noise distribution is $d$-correlated if for any two sets $S$ and $T$ such that $|S \backslash T| + |T \backslash S| > d$, the noise is applied independently to $S$ and to $T$. The noise multipliers can be arbitrarily correlated when $S$ and $T$ are similar in the sense that $|S \backslash T| + |T \backslash S| \leq d$. We notice that our algorithms can be naturally extended to $d$-correlated noise for $d \in O(1)$. In particular, to adapt Algorithm 2 to deal with $d$-correlated noise, we need to arbitrarily split $N$ into sets $T_1, \ldots, T_{\lfloor \frac{N}{d+1} \rfloor}$ and define the smoothing surrogate function as $h(S) = \frac{1}{L} \sum_{l=1}^{L} f(S \cup T_l)$ to replace the original surrogate.

## 7  Conclusions

In this work, we study the problem of constrained monotone submodular maximization with noisy oracles. We design a unified local search framework that allows for inaccuracy in the objective function. Under this framework, we construct several smoothing surrogate functions to average the noise out. For cardinality constraints, the local search framework results in algorithms that achieve a $\left(1 - \frac{1}{e} - O(\varepsilon)\right)$-approximation with $\text{Poly}\left(n, \frac{1}{\varepsilon}\right)$ query complexity. Moreover, for general matroid constraints, the framework obtains an approximation ratio arbitrarily close to $\left(1 - \frac{1}{e}\right)/2$, which is the first constant approximation result to our knowledge.

There are many directions in which this work could be extended. For submodular maximization with noisy oracles under general matroid constraints, there is a gap between the approximation ratio of $(1 - 1/e)/2 - O(\varepsilon)$ provided in this paper and impossibility results [5, 20]. The first open question is to close this gap. In addition, it would be interesting to consider more complex constraints under noise including knapsack constraints [4]. It is meaningful to investigate submodular maximization with correlated noise. Moreover, it is also worthwhile to investigate the robustness of other submodular optimization approaches, e.g., multi-linear extension [25].

## Acknowledgements

We are grateful to Dr. Bei Xiaohui for his good counsel and valuable comments on the manuscript.

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
