## A  The Invalid Result for General Matroid Constraints in [20]

In this section, we recall Algorithm 4 ([20, Algorithm 2]) and its analysis Theorem A.1 ([20, Theorem 4.2]) proposed by Singer and Hassidim [20] for monotone submodular maximization with noisy oracles under general matroid constraints. We argue that Algorithm 4 fails to obtain the approximation guarantee claimed in Theorem A.1.

For a set $S \subseteq N$, a bundle $\boldsymbol{b} \subseteq N$ and an intersection of matroids $\mathcal{F}$, the *mean value*, *noisy mean value* and *mean marginal contribution* of $\boldsymbol{b}$ given $S$ are, respectively:

$$F(S \cup \boldsymbol{b}) := \mathbb{E}_{\boldsymbol{z} \sim \mathcal{B}_S(\boldsymbol{b})} \left[ f(S \cup \boldsymbol{z}) \right]$$

$$\tilde{F}(S \cup \boldsymbol{b}) := \mathbb{E}_{\boldsymbol{z} \sim \mathcal{B}_S(\boldsymbol{b})} \left[ \tilde{f}(S \cup \boldsymbol{z}) \right]$$

$$F_S(\boldsymbol{x}) := \mathbb{E}_{\boldsymbol{z} \sim \mathcal{B}_S(\boldsymbol{b})} \left[ f_S(\boldsymbol{b}) \right]$$

where $\mathcal{B}_S(\boldsymbol{b})$ represents the ball around $\boldsymbol{b}$, i.e., $\mathcal{B}_S(\boldsymbol{b}) = \{\boldsymbol{b} - x_i + x_j \in \mathcal{F} : x_i \in \boldsymbol{b}, x_j \notin S \cup \boldsymbol{x}\}$.

---

**Algorithm 4:** SM-MATROID-GREEDY

---

**Input :** intersection of matroids $\mathcal{F}$, precision $\varepsilon > 0$, $c \leftarrow \frac{56}{\varepsilon}$

1  $S \leftarrow \varnothing, X \leftarrow N$
2  **while** $X \neq S$ **do**
3  $\quad X \leftarrow X \backslash \{\boldsymbol{x} : S \cup \boldsymbol{x} \notin \mathcal{F}\}$
4  $\quad \boldsymbol{x} \leftarrow \arg\max_{\boldsymbol{b}:|\boldsymbol{b}|=c} \tilde{F}(S \cup \boldsymbol{b})$
5  $\quad \hat{\boldsymbol{x}} \leftarrow \arg\max_{\boldsymbol{z} \in \mathcal{B}_S(\boldsymbol{x})} \tilde{f}(S \cup \boldsymbol{z})$
6  $\quad S \leftarrow S \cup \hat{\boldsymbol{x}}$
7  **return** $S$

---

Algorithm 4 is a variant of the standard greedy algorithm, which at every iteration adds a bundle $\hat{\boldsymbol{x}}$ of size $c = \Theta(1/\varepsilon)$ instead of a single element. In each iteration, the algorithm updates the set $X$ of candidate elements (Line 3) in order to obtain a feasible set $S \in \mathcal{F}$. Then it selects a bundle $\boldsymbol{x} \in \arg\max_{\boldsymbol{b}:|\boldsymbol{b}|=c} \tilde{F}(S \cup \boldsymbol{b})$ with the largest noisy mean value $\tilde{F}(S \cup \boldsymbol{b})$ (Line 4). Finally, Algorithm 4 evaluates all possible bundles $\boldsymbol{z} \in \mathcal{B}_S(\boldsymbol{x})$ in the ball around $\boldsymbol{x}$ identified at last step and incorporates the one whose noisy value $\tilde{f}(S \cup \boldsymbol{z})$ is largest (Line 5 and 6).

Singer and Hassidim [20] claim that Algorithm 4 can achieve the following approximation performance.

**Theorem A.1** (**Wrong theorem [20, Theorem 4.2]**)**.** *Let $\mathcal{F}$ denote the intersection of $P \geq 1$ matroids with rank $r \in \Omega\left(\frac{1}{\varepsilon^2}\right) \cap \sqrt{\log n}$ on the ground set $N$, and $f : 2^N \to \mathbb{R}$ be a non-negative monotone submodular function. Then with probability $1 - o(1)$ the* SM-MATROID-GREEDY *algorithm returns a set $S \in \mathcal{F}$ s.t.:*

$$f(S) \geq \frac{1 - \varepsilon}{P + 1}.$$

Now we argue that Theorem A.1 does not hold even under a single matroid constraint. Singer and Hassidim [20] employed [20, Lemma 3.4] to prove Theorem A.1. However, this lemma cannot be generalized beyond cardinality constraints to matroid constraints. This is because [20, Lemma 3.4] is based on the following fact: for any bundle $\boldsymbol{b}$ of size $1/\varepsilon$,

$$F_S(\boldsymbol{b}) \geq (1 - \varepsilon) f_S(\boldsymbol{b}). \qquad \text{([20, Lemma 2.2])}$$

This fact results in [20, Corollary 2.3], which appears as the premise of [20, Claim 3.1] to prove [20, Lemma 3.4]. Under a cardinality constraint, the ball $\mathcal{B}_S(\boldsymbol{b})$ contains all the neighbors of a bundle $\boldsymbol{b}$, which differ from $\boldsymbol{b}$ by only one element. A bundle $\boldsymbol{b}$ with good margin is likely to be surrounded by neighbors in $\mathcal{B}_S(\boldsymbol{b})$ which also have large margins on average, and the fact that $F_S(\boldsymbol{b}) \geq (1-\varepsilon) f_S(\boldsymbol{b})$ naturally holds. However, when we consider a matroid that restricts feasible neighbors of $\boldsymbol{b}$ to be those with small margins only, $F_S(\boldsymbol{b})$ can be far less than $f_S(\boldsymbol{b})$.

We provide a concrete example to show that Algorithm 4 cannot obtain any with-high-probability constant approximation under a matroid constraint.

**Claim A.2.** *For any constant $m$, there exists a partition matroid $\mathcal{M}$ with rank $r \in \Omega\left(\frac{1}{\varepsilon^2}\right) \cap O(\sqrt{\log n})$ for which Algorithm 4 fails to achieve an approximate ratio better than $1/m$ with probability at least $1/3$.*

*Proof.* Our plan is to construct a partition matroid for which Algorithm 4 performs badly.

**Definition A.3 (Partition matroid).** A matroid $\mathcal{M} = (N, \mathcal{I}(\mathcal{M}))$ is a partition matroid if $N$ is partitioned into $k$ disjoint sets $C_1, C_2, \ldots, C_k$ and

$$\mathcal{I}(\mathcal{M}) = \{S \subseteq N : |C_i \cap S| \leq d_i \text{ for } i = 1, 2, \ldots, k\}.$$

Let there be $r$ disjoint sets, where $r \in \Omega\left(\frac{1}{\varepsilon^2}\right) \cap O(\sqrt{\log n})$. For all $i \in [r-1]$, $C_i$ contains only one element with value 0. The set $C_r$ with $|C_r| = n - r + 1$ contains most of the elements. There is a special element $e^\star$ in $C_r$ with $f(e^\star) = m$, and other elements in $C_r$ are all attributed with value 1. An independent set of matroid $\mathcal{M}$ contains at most one element from each disjoint set. That is, $d_1 = d_2 = \cdots = d_r = 1$. The noise distribution will return $m$ with probability $\delta = \frac{1}{2(n-r)}$ and 1 otherwise.

Now we apply Algorithm 4 to maximize $f$ under this matroid constraint with a noisy oracle. Since the elements in $C_i$ ($i \in [r-1]$) are all with value 0, we focus on the bundles $\boldsymbol{b}$ that consist of an element in $C_r$ and some other elements $\boldsymbol{b}_{-r}$. Suppose Algorithm 4 selects one of them as $\boldsymbol{x}$ at Line 4. Feasible bundles in $\mathcal{B}_S(\boldsymbol{x})$ containing $e^\star$ can only be obtained by changing one of the elements in $\boldsymbol{x}_{-r}$. Thus there are at most $(c-1)(r-c)$ such bundles. With probability at least $(1-\delta)^{(c-1)(r-c)}$, the noise multipliers on such bundles are all 1.

On the other hand, at least $n - r$ bundles in $\mathcal{B}_S(\boldsymbol{x})$ do not include $e^\star$. With probability at least $1 - (1-\delta)^{n-r}$, there exists one of these bundles with noise multiplier $m$. Thus Algorithm 4 selects a bundle that does not include $e^\star$ at Line 5 with probability at least

$$(1-\delta)^{(c-1)(r-c)}\left[1 - (1-\delta)^{n-r}\right] \geq \left(1 - \frac{(c-1)(r-c)}{2(n-r)}\right)\left(1 - \frac{1}{\sqrt{e}}\right).$$

When $n$ is sufficiently large, the probability above is at least $1/3$. $\qquad\square$

By the proof of Claim A.2, we have an intuition that the problem of submodular maximization under a partition matroid with any rank may degenerate to that under 1-cardinality constraint. The case of $r = 1$ is rather difficult for Problem 2.6, and the only known result is an algorithm by [20] which achieves a $1/2$-approximation guarantee in expectation for this case. To the best of our knowledge, there is no with-high-probability result before.

# B   More Related Work

Besides cardinality and single matroid constraints, more complex constraints have also been considered in the context of submodular optimization before. Fisher et al. [8] present an algorithm achieving a $1/(k+1)$-approximation for monotone submodular maximization under $k$ matroid constraints. Lee et al. [17] subsequently improve the approximation guarantee to $1/(k+\varepsilon)$ for $k > 2$. Sviridenko [22] provides a $\left(1 - \frac{1}{e}\right)$-approximation under knapsack constraints. With the multilinear relaxation technique, Chekuri et al. [4] obtain a $0.38/k$-approximation for maximizing a monotone submodular function subject to $k$ matroids and a constant number of knapsack constraints.

# C   Proof of Theorem 3.3: Analysis of Algorithm 1

We first analyze the query complexity to $\widehat{\varphi_h}$ and then prove the approximate ratio together with the success probability.

**Query complexity of Algorithm 1.** For any $i \in [r]$, define $\mathcal{U}_i := \{U_{i-1} + e \in \mathcal{I}(\mathcal{M}) : e \in N \setminus U_{i-1}\}$ to be the collection of subsets considered in Line 4. For any $i = 0, 1 \cdots, I - 1$, define

$\mathcal{U}'_i := \{S'_i \in \mathcal{I}(\mathcal{M})\}$ to be the collection of subsets considered in Line 10 in interation $i$. Let $\mathcal{U} = \left( \bigcup_{i \in [r]} \mathcal{U}_i \right) \cup \left( \bigcup_{0 \le i \le I-1} \mathcal{U}'_i \right)$. Since each $|\mathcal{U}_i| \le n$ and each $|\mathcal{U}'_i| \le rn$, we have

$$|\mathcal{U}| \le (I+1)rn.$$

Thus the total query complexity is less than $(I+1)rn$.

**Approximate ratio and success probability of Algorithm 1.** We say that event $\mathcal{E}_g$ happens if for all sets $A \in \mathcal{U}$, the following inequality holds:

$$\mathbb{P}\left[ |\widehat{\varphi_h}(A) - \varphi_h(A)| > \alpha \max_{S \in \mathcal{I}(\mathcal{M})} \varphi_h(S) \right] \le \delta.$$

By the above argument, we know that the total number of calls to $\widehat{\varphi_h}$ is at most $(I+1)rn$. Thus

$$\mathbb{P}[\mathcal{E}_g] \ge 1 - (I+1)rn\delta.$$

In the remaining proof, we assume $\mathcal{E}_g$ holds. Firstly we prove the following lemma which analyzes the performance of the greedy initial solution $S_0$ in Line 7 of Algorithm 1.

**Lemma C.1 (Approximation ratio of $S_0$).** *Conditioned on $\mathcal{E}_g$, we have*

$$\varphi_h(S_0) \ge \frac{1 - 2r\alpha}{2} \max_{S \in \mathcal{I}(\mathcal{M})} \varphi_h(S).$$

*Proof.* We first introduce the following well-known fact.

**Fact C.2 (Brualdi's lemma).** *Suppose $A$, $B$ are two bases in a matroid. There is a bijection $\pi : A \to B$ such that for all $a \in A$, $A - a + \pi(a)$ is a base. Furthermore, $\pi$ is the identity on $A \cap B$.*

Let $O^\star \in \arg\max_{S \in \mathcal{I}(\mathcal{M})} \varphi_h(S)$ be a set on which $\varphi_h$ attains its maximum value. For $O^\star$ and $U_r$, we index the elements of $O^\star$ as $\{o_1, \dots, o_r\}$ according to the Brualdi's lemma such that $o_i = \pi(u_i)$ with $U_{i-1} + o_i \in \mathcal{I}(\mathcal{M})$. From how we choose $u_i$ in Line 4 of Algorithm 1, we have

$$\widehat{\varphi_h}(U_{i-1} + u_i) \ge \widehat{\varphi_h}(U_{i-1} + o_i) \tag{2}$$

for all $i \in [r]$. To utilize monotonicity and submodularity of $\varphi_h$, we translate the inequality above into that of $\varphi_h$:

$$
\begin{aligned}
\varphi_h(U_{i-1} + u_i) + \alpha\varphi_h(O^\star) &\ge \widehat{\varphi_h}(U_{i-1} + u_i) && \text{(event } \mathcal{E}_g) \\
&\ge \widehat{\varphi_h}(U_{i-1} + o_i) && \text{(Ineq. (2))} \\
&\ge \varphi_h(U_{i-1} + o_i) - \alpha\varphi_h(O^\star) && \text{(event } \mathcal{E}_g).
\end{aligned}
$$

That is,

$$\varphi_h(U_{i-1} + u_i) \ge \varphi_h(U_{i-1} + o_i) - 2\alpha \cdot \varphi_h(O^\star).$$

Subtracting $\varphi_h(U_{i-1})$ from each side gives

$$\varphi_h(U_{i-1} + u_i) - \varphi_h(U_{i-1}) \ge [\varphi_h(U_{i-1} + o_i) - \varphi_h(U_{i-1})] - 2\alpha \cdot \varphi_h(O^\star)$$

for all $i \in [r]$. Summing these $r$ inequalities, we obtain

$$\varphi_h(S_0) \ge \sum_{i=1}^{r-1} [\varphi_h(U_{i-1} + o_i) - \varphi_h(U_{i-1})] - 2r\alpha \cdot \varphi_h(O^\star). \tag{3}$$

From submodularity of $\varphi_h$, we have

$$\sum_{i=1}^{r-1} [\varphi_h(U_{i-1} + o_i) - \varphi_h(U_{i-1})] \ge \sum_{i=1}^{r-1} [\varphi_h(S_0 + o_i) - \varphi_h(S_0)] \ge \varphi_h(S_0 \cup O^\star) - \varphi_h(S_0). \tag{4}$$

Combining (3) and (4) gives

$$2\varphi_h(S_0) \ge \varphi_h(S_0 \cup O^\star) - 2r\alpha \cdot \varphi_h(O^\star) \ge (1 - 2r\alpha)\varphi_h(O^\star).$$

This completes the proof. $\qquad\square$

**Lemma C.3** (**Maximum iterations of the local search procedure in Algorithm 1**). *Conditioned on event $\mathcal{E}_g$, the local search procedure in Algorithm 1 terminates within $I = \log_{1+\Delta}\left(\frac{2(1+\alpha)}{1-2(r+1)\alpha}\right)$ iterations.*

*Proof.* Let $O^\star \in \arg\max_{S \in \mathcal{I}(\mathcal{M})} \varphi_h(S)$ be a set on which $\varphi_h$ attains its maximum value and $O' \in \arg\max_{S \in \mathcal{U}} \widehat{\varphi_h}(S)$ be the set with largest value of $\widehat{\varphi_h}$. By Lemma C.1, we have

$$
\begin{aligned}
\widehat{\varphi_h}(S_0) &\geq \varphi_h(S_0) - \alpha\varphi_h(O^\star) \\
&\geq \frac{1-2(r+1)\alpha}{2}\varphi_h(O^*) &&\text{(Lemma C.1)} \\
&\geq \frac{1-2(r+1)\alpha}{2(1+\alpha)}\varphi_h(O') + \frac{(1-2(r+1)\alpha)\alpha}{2(1+\alpha)}\varphi_h(O^*) &&\text{(optimality of } O^*\text{)} \\
&\geq \frac{1-2(r+1)\alpha}{2(1+\alpha)}\left(\widehat{\varphi_h}(O') - \alpha\varphi_h(O^*)\right) + \frac{(1-2(r+1)\alpha)\alpha}{2(1+\alpha)}\varphi_h(O^*) \\
&= \frac{1-2(r+1)\alpha}{2(1+\alpha)}\widehat{\varphi_h}(O').
\end{aligned}
$$

Since every local search step increases the function value $\widehat{\varphi_h}$ by $1 + \Delta$, Algorithm 1 searches for at most $I = \log_{1+\Delta}\left(\frac{2(1+\alpha)}{1-2(r+1)\alpha}\right)$ iterations. $\square$

Conditioned on $\mathcal{E}_g$, we finally obtain the approximation ratio of $S_I$ by the locally optimality of $S_I$ by Theorem 1 and Lemma 4 in [7].

**Lemma C.4** ([7, Theorem 1]). *Let $A = \{a_1, \cdots, a_r\}$ and $B = \{b_1, \cdots, b_r\}$ be any two bases of $\mathcal{M}$. Further suppose that we index the elements of $B$ so that $b_i = \pi(a_i)$, where $\pi : A \to B$ is the bijection guaranteed by Fact C.2 (Brualdi's lemma). Then,*

$$
\sum_{i=1}^{r}[\varphi_h(A - a_i + b_i) - \varphi_h(A)] \geq h(B) - \frac{e}{e-1}h(A).
$$

**Lemma C.5** ([7, Lemma 4]). *For all $A \subseteq N$,*

$$
h(A) \leq \varphi_h(A) \leq \frac{e}{e-1}H_{|A|}h(A),
$$

*where $H_{|A|} = \sum\limits_{i=1}^{|A|}\frac{1}{i}$.*

Let $O \in \arg\max_{S \in \mathcal{I}(\mathcal{M})} h(S)$ and $O^\star \in \arg\max_{S \in \mathcal{I}(\mathcal{M})} \varphi_h(S)$. By Fact C.2, we index the elements of $S_I$ and $O$ by $S_I = \{s_1, \cdots, s_r\}$ and $O = \{o_1, \cdots, o_r\}$ such that $S_I - s_i + o_i \in \mathcal{I}(\mathcal{M})$ holds for all $i \in [r]$. We firstly transfer $S_I$'s the locally optimality w.r.t. $\widehat{\varphi_h}$ to the locally optimality w.r.t. $\varphi_h$. For all $s_i$ and $o_i$, we have

$$
\begin{aligned}
\varphi_h(S_I - s_i + o_i) - \alpha\varphi_h(O^*) &\leq \widehat{\varphi_h}(S_I - s_i + o_i) &&\text{(event } \mathcal{E}_g\text{)} \\
&\leq (1+\Delta)\widehat{\varphi_h}(S_I) &&\text{(Lemma C.3)} \\
&\leq (1+\Delta)\left(\varphi_h(S_I) + \alpha\varphi_h(O^\star)\right) &&\text{(event } \mathcal{E}_g\text{)}
\end{aligned}
$$

which implies

$$
\varphi_h(S_I - s_i + o_i) - \varphi_h(S_I) \leq \Delta\varphi_h(S_I) + (2+\Delta)\alpha\varphi_h(O^\star).
$$

Summing the resulting $r$ inequalities gives

$$
\begin{aligned}
\sum_{i=1}^{r}[\varphi_h(S_I - s_i + o_i) - \varphi_h(S_I)] &\leq r\Delta\varphi_h(S_I) + r(2+\Delta)\alpha\varphi_h(O^\star) \\
&\leq rH_r\Delta h(S_I) + r(2+\Delta)\alpha H_r h(O^\star) &&\text{(Lemma C.5)}
\end{aligned}
$$

$$\leq \; rH_r\left((2+\Delta)\alpha + \Delta\right)h(O). \qquad\qquad \text{(optimality of } O)$$

Combine the Lemma C.4, we have

$$h(O) - \frac{e}{e-1}h(S_I) \leq \sum_{i=1}^{r}[\varphi_h(S_I - s_i + o_i) - \varphi_h(S_I)] \leq rH_r\left((2+\Delta)\alpha + \Delta\right)h(O).$$

Thus

$$h(S_I) \; \geq \; \left(1 - \frac{1}{e}\right)\left(1 - rH_r\left((2+\Delta)\alpha + \Delta\right)\right)h(O)$$

$$\geq \; \left(1 - \frac{1}{e}\right)\left(1 - r\left(\ln r + 1\right)\left((2+\Delta)\alpha + \Delta\right)\right)h(O).$$

This completes the proof of Theorem 3.3.

## D  Missing Proofs in Subsection 4.1

In this appendix, we provide the proofs that we omitted from Subsection 4.1.

### D.1  Proof of Lemma 4.3

For any set $S \subseteq N$ and $x \notin S$, by Definition 4.2 we have

$$h(S+x) - h(S) = \frac{1}{n}\sum_{e\in N}\left[f(S+x+e) - f(S+e)\right]. \qquad\qquad (5)$$

This immediately implies that $h$ is monotone, since the monotonicity of $f$ implies that each term $f(S+x+e) - f(S+e)$ is non-negative. Next, suppose that $T \subseteq S$ and $x \notin S$. Then

$$h(T+x) - h(T) = \frac{1}{n}\sum_{e\in N}\left[f(T+x+e) - f(T+e)\right] \qquad\qquad \text{(Eq. (5))}$$

$$\geq \frac{1}{n}\sum_{e\in N}\left[f(S+x+e) - f(S+e)\right] \qquad\qquad \text{(submodularity of } f)$$

$$= h(S+x) - h(S). \qquad\qquad \text{(Eq. (5))}$$

Thus, $h$ is submodular.

### D.2  Proof of Lemma 4.4

Here we illustrate intuitively how to obtain the value oracle $\mathcal{O}$ to an $(\alpha, \delta, \mathcal{I}(r-1))$-approximation $\widehat{\varphi_h}$ of $\varphi_h$. When $n$ is large enough, the generalized exponential tail allows the difference between $\varphi_h$ and its noisy analogue $\widetilde{\varphi_h}$ to be averaged out. Thus we can obtain an approximation $\widehat{\varphi_h}$ of $\varphi_h$ by approximating $\widetilde{\varphi_h}$. Since evaluating $\widetilde{\varphi_h}$ exactly will require an exponential number of value queries to $\tilde{f}$, we use a random sampling procedure (Algorithm 5) to estimate it. With only polynomial queries to $\tilde{f}$, $\widetilde{\varphi_h}$ can be approximated to the desired accuracy.

To prove Lemma 4.4, we first show some bounds for coefficients $\tau_A(T)$. With these bounds, we will then prove that when $n$ is sufficiently large, $\varphi_h$ and its noisy analogue $\widetilde{\varphi_h}$ are close to each other by Bernstein's inequality for sub-exponential variables. Finally, we use Hoeffding's inequality to show that with enough samples, $\widehat{\varphi_h}$ is close to $\widetilde{\varphi_h}$ and therefore a good approximation of $\varphi_h$.

For a set $A \subseteq N$, we rewrite the auxiliary function $\varphi_h(A)$ as a linear combination of $f(T)$ according to Definition 3.1 and 4.2 and obtain

$$\varphi_h(A) = \frac{1}{n}\sum_{T\subseteq A}\left[(m_{|A|-1,|T|-1} + m_{|A|-1,|T|-2})|T|\cdot f(T) + \sum_{e\in N\setminus A}m_{|A|-1,|T|-1}\cdot f(T+e)\right]. \quad (6)$$

We define the averaging set of $\varphi_h(A)$ as

$$\mathcal{L}_A = \left\{T \in 2^N : T \subseteq A, \text{ or } \exists\, S \subseteq A \text{ and } e \in N\setminus A \text{ s.t. } S + e = T\right\}.$$

For any $T \in \mathcal{L}_A$, we denote by $\tau_A(T)$ the linear coefficient of $f(T)$. Then we can write $\varphi_h(A)$ as $\sum_{T \in \mathcal{L}_A} \tau_A(T) f(T)$.

**Lemma D.1 (Useful bounds of the linear coefficients).** *For any $A \subseteq N$ with $|A| \leq \sqrt{n}$, we have*

$$(1) \sum_{T \in \mathcal{L}_A} \tau_A^2(T) \leq \frac{12}{n|A|}; \qquad (2) \max_{T \in \mathcal{L}_A} \tau_A(T) \leq \frac{4}{n}; \qquad (3) \sum_{T \in \mathcal{L}_A} \tau_A(T) \leq 2\left(\ln|A| + 2\right).$$

*Proof.* For notation convenience, we let $a = |A|$ and $t = |T|$ in this proof. For any $A \subseteq N$, we have

$$\sum_{T \in \mathcal{L}_A} \tau_A^2(T)$$

$$= \frac{1}{n^2} \sum_{t=0}^{a} \binom{a}{t} \left[ (m_{a-1,t-1} + m_{a-1,t-2})^2 \cdot t^2 + m_{a-1,t-1}^2 \cdot (n-a) \right] \qquad \text{(Eq. (6))}$$

$$= \frac{1}{n^2} \left\{ \binom{a}{1} \left( \int_0^1 \frac{e^p}{e-1}(1-p)^{a-1} dp \right)^2 (n-a+1) \right. \qquad \text{(Definition 3.1)}$$

$$\left. + \sum_{t=2}^{a} \binom{a}{t} \left[ \left( \int_0^1 \frac{e^p}{e-1} p^{t-2}(1-p)^{a-t} dp \right)^2 t^2 + \left( \int_0^1 \frac{e^p}{e-1} p^{t-1}(1-p)^{a-t} dp \right)^2 (n-a) \right] \right\}$$

$$\leq \frac{e^2}{(e-1)^2 n^2} \left\{ \sum_{t=2}^{a} \left( \frac{at}{(t-1)^2} + \frac{n-a}{at} \right) \frac{\Gamma(t)\Gamma(a-t+1)}{\Gamma(a)} + \frac{n-a+1}{a} \right\} \qquad (7)$$

$$\leq \frac{e^2}{(e-1)^2 n^2} \left\{ \frac{1}{a-1} \sum_{t=2}^{a} \left( 2a + \frac{n-a}{a} \right) \frac{1}{\binom{a-2}{t-2}} + \frac{n-a+1}{a} \right\} \qquad (\tfrac{t}{t-1} \leq 2 \text{ for } t \geq 2)$$

$$< \frac{6}{n^2} \left( a + \frac{n}{a} \right), \qquad (\tfrac{e^2}{(e-1)^2} \leq 3 \text{ and } \binom{a-2}{t-2} \geq 1)$$

where we use the fact that $e^p \leq e$ for $p \in [0, 1]$ and a property of beta function that

$$B(x, y) = \int_0^1 p^{x-1}(1-p)^{y-1} dp = \frac{\Gamma(x)\Gamma(y)}{\Gamma(x+y)} \qquad (8)$$

to obtain Ineq. (7). In particular, if $|A| \leq \sqrt{n}$, we have $\sum_{T \in \mathcal{L}_A} \tau_A^2(T) \leq 12/(n|A|)$.

Next we show that $\max_{T \in \mathcal{L}_A} \tau_A(T) \leq 4/n$. For any set $T \subseteq A$ with $|T| \geq 2$, we have

$$\tau_A(T) = \frac{1}{n}(m_{a-1,t-1} + m_{a-1,t-2}) \cdot t \qquad \text{(Eq. (6))}$$

$$= \frac{t}{n} \int_0^1 \frac{e^p}{e-1} p^{t-2}(1-p)^{a-t} dp \qquad \text{(Definition 3.1)}$$

$$\leq \frac{e}{e-1} \cdot \frac{t}{n} \cdot \frac{(t-2)!(a-t)!}{(a-1)!} \qquad (p < 1 \text{ and Eq. (8))}$$

$$\leq \frac{2}{n} \cdot \frac{t}{a-1} \leq \frac{4}{n}. \qquad (\tfrac{e}{e-1} \leq 2, \binom{a-2}{t-2} \geq 1 \text{ and } \tfrac{t}{a-1} \leq 2)$$

Similarly, for a set $T$ that only contains a single element in $A$,

$$\tau_A(T) = \frac{m_{a-1,0}}{n} = \frac{1}{n} \int_0^1 \frac{e^p}{e-1}(1-p)^{a-1} dp \leq \frac{2}{an}.$$

For any $T \in \mathcal{L}_A \setminus 2^A$, we have

$$\tau_A(T) = \frac{1}{n} \cdot m_{a-1,t-2} \qquad \text{(Eq. (6))}$$

$$= \frac{1}{n} \int_0^1 \frac{e^p}{e-1} p^{t-2}(1-p)^{a-t+1} dp \qquad \text{(Definition 3.1)}$$

$$\leq \frac{e}{(e-1)n} \cdot \frac{1}{\binom{a}{t-2}} \cdot \frac{1}{a-t+2} \leq \frac{2}{n}. \qquad \text{(Eq. (8) and } 2 \leq t \leq a+1)$$

Combining these three parts gives $\max_{T \in \mathcal{L}_A} \tau_A(T) \leq 4/n$. Finally we can bound $\sum_{T \in \mathcal{L}_A} \tau_A(T)$ as below:

$$\sum_{T \in \mathcal{L}_A} \tau_A(T)$$

$$= \frac{1}{n} \sum_{t=0}^{a} \binom{a}{t} \left[ (m_{a-1,t-1} + m_{a-1,t-2})t + m_{a-1,t-1}(n-a) \right] \qquad \text{(Eq. (6))}$$

$$= \frac{1}{n} \left\{ \sum_{t=2}^{a} \binom{a}{t} \left[ t \int_0^1 \frac{e^p}{e-1} p^{t-2}(1-p)^{a-t} dp + (n-a) \int_0^1 \frac{e^p}{e-1} p^{t-1}(1-p)^{a-t} dp \right] \right.$$

$$\left. + \binom{a}{1}(n-a+1) \int_0^1 \frac{e^p}{e-1}(1-p)^{a-1} dp \right\} \qquad \text{(Definition 3.1)}$$

$$\leq \frac{e}{(e-1)n} \cdot \left[ \sum_{t=2}^{a} \left( \frac{a}{t-1} + \frac{n-a}{t} \right) + (n-a+1) \right] \qquad (p < 1)$$

$$\leq 2(\ln a + 2). \qquad (\tfrac{e}{e-1} \leq 2 \text{ and } \sum_{t=2}^{a} \tfrac{1}{t-1} \leq \ln a + 1)$$

$\square$

We now use concentration results of sub-exponential distributions to bound the difference between $\varphi_h$ and its noisy analogue $\widetilde{\varphi_h}$.

**Definition D.2 (Sub-exponential distribution).** The sub-exponential norm of $X \in \mathbb{R}$ is

$$\|X\|_{\psi_1} = \inf\{t > 0 : \mathbb{E}\exp(|X|/t) \leq 2\}.$$

If $\|X\|_{\psi_1}$ is finite, we say that $X$ is sub-exponential.

**Claim D.3.** *Let $\xi$ be a random variable with a generalized exponential tail distribution $\mathcal{D}$. Then $\xi - 1$ is sub-exponential.*

*Proof.* If the support of $\mathcal{D}$ is bounded by some constant $b$, then let $\kappa = (b+1)/\ln 2$, and we have $\|\xi - 1\|_{\psi_1} \leq \kappa$. For the case where $\mathcal{D}$ does not have bounded support, recall that the probability density function $\rho(x)$ of $\mathcal{D}$ is $\exp(-\sum_i c_i x^{\gamma_i})$. When

$$x \geq \max\left\{ 1, \left( \frac{2\sum_{i=1} |c_i|}{c_0} \right)^{\frac{1}{\gamma_0 - \gamma_1}} \right\},$$

the term $\frac{1}{2} c_0 x^{\gamma_0}$ dominates the rest of the terms, and thus $\rho(x) \leq e^{-\frac{1}{2}c_0 x^{\gamma_0}} \leq e^{-\frac{1}{2}c_0 x}$. Let

$$\kappa = \max\left\{ 2, \frac{2}{c_0} \cdot \ln\left( \frac{8}{c_0} \right) + 1, \ln\left( \frac{2}{3} \right) \cdot \left( \frac{2\sum_{i=1} |c_i|}{c_0} \right)^{\frac{1}{\gamma_0 - \gamma_1}} \right\}.$$

It is straightforward to verify that $\mathbb{E}\left[ \exp\left( \frac{|\xi-1|}{\kappa} \right) \right] \leq 2$. Hence $\xi - 1$ is a sub-exponential variable with a constant sub-exponential norm $\|\xi - 1\|_{\psi_1} \leq \kappa$. $\square$

For sub-exponential random variables, we have the following Bernstein's inequality.

**Lemma D.4** (**Bernstein's inequality for sub-exponential variables [24]**). *Let $X_1, \ldots, X_m$ be independent, mean zero, sub-exponential random variables, and $a = (a_1, \ldots, a_m) \in \mathbb{R}^m$. Then, for every $t > 0$, we have*

$$\mathbb{P}\left[ \left| \sum_{i=1}^{m} a_i X_i \right| \geq t \right] \leq 2\exp\left[ -c\min\left( \frac{t^2}{K^2 \|a\|_2^2}, \frac{t}{K \|a\|_\infty} \right) \right], \qquad (9)$$

*where $K = \max_i \|X_i\|_{\psi_1}$ and $c$ is a constant.*

With Bernstein's inequality, we can bound the difference between $\varphi_h$ and $\widetilde{\varphi_h}$ by the following lemma.

**Lemma D.5 (Difference between $\varphi_h$ and $\widetilde{\varphi_h}$).** *Suppose that $n \geq 192\kappa^2 c^{-1} \cdot \alpha^{-2} \ln(4\delta^{-1})$, where $\kappa$ denotes the sub-exponential norm of the noise multiplier and $c$ is the constant in Ineq. (9). For any set $A \in \mathcal{I}(r-1)$, we have*

$$\mathbb{P}\left[|\widetilde{\varphi_h}(A) - \varphi_h(A)| > \frac{\alpha}{2} \cdot \max_{S \in \mathcal{I}(r-1)} \varphi_h(S)\right] \leq \frac{\delta}{2}.$$

*Proof.* Let us fix a set $A \in \mathcal{I}(r-1)$. From monotonicity and submodularity of $f$, for any $T \in \mathcal{L}_A$,

$$f(T) \leq f(A) + \max_{e \in N} f(e) \leq 2 \cdot \max_{S \in \mathcal{I}(r-1)} f(S) \leq 2 \cdot \max_{S \in \mathcal{I}(r-1)} \varphi_h(S). \tag{10}$$

Let $X_T = \xi(T) - 1$ and $a_T = \tau_A(T)f(T)$. By Lemma D.1, we have

$$\sum_{T \in \mathcal{L}_A} (\tau_A(T)f(T))^2 \leq 4 \left(\sum_{T \in \mathcal{E}_A} \tau_A^2(T)\right) \left(\max_{S \in \mathcal{I}(r-1)} \varphi_h(S)\right)^2 \leq \frac{48}{|A|n} \cdot \left(\max_{S \in \mathcal{I}(r-1)} \varphi_h(S)\right)^2$$

and

$$\max_{T \in \mathcal{L}_A} \tau_A(T)f(T) \leq 2 \max_{T \in \mathcal{L}_A} \tau_A(T) \cdot \max_{S \in \mathcal{I}(r-1)} \varphi_h(S) \leq \frac{8}{n} \cdot \max_{S \in \mathcal{I}(r-1)} \varphi_h(S).$$

Since we have $\alpha|A| < \varepsilon/(4H_{r-1}) < 12\kappa$, applying Lemma D.4 gives

$$\mathbb{P}\left[|\widetilde{\varphi_h}(A) - \varphi_h(A)| \geq \frac{\alpha}{2} \cdot \max_{S \in \mathcal{I}(r-1)} \varphi_h(S)\right] \leq 2\exp\left(-c \cdot \frac{n\alpha^2|A|}{192\kappa^2}\right).$$

By assumption that $n \geq 192\kappa^2 c^{-1} \cdot \varepsilon_0^{-2} \ln(4\delta^{-1})$, the probability above is at most $\delta/2$. □

The random sampling procedure that we use to estimate $\widetilde{\varphi_h}$ is presented as Algorithm 5.

---

**Algorithm 5:** Approximation of $\varphi_h$

**Input :** Noisy oracle $\tilde{f}$, a set $A \in \mathcal{I}(r-1)$, number of samples $M$.
1 Construct a distribution $\nu(T)$ on set $\mathcal{L}_A$ such that $\nu(T) = \tau_A(T)/\left(\sum_{T' \in \mathcal{L}_A} \tau_A(T')\right)$
2 Sample $M$ sets $T_1, \ldots, T_M$ in $\mathcal{L}_A$ according to the distribution $\nu(T)$
3 $\widehat{\varphi_h}(A) \leftarrow \left(\sum_{T' \in \mathcal{L}_A} \tau_A(T')\right) \cdot \frac{1}{M} \sum_{j=1}^{M} \tilde{f}(T_j)$
4 **return** $\widehat{\varphi_h}(A)$

---

We plan to use Hoeffding's inequality to show that $\widehat{\varphi_h}(A)$ and $\widetilde{\varphi_h}(A)$ are very close to each other, and thus $\widehat{\varphi_h}$ is a $(\alpha, \delta, \mathcal{I}(r-1))$-approximation of $\varphi_h$.

**Lemma D.6 (Hoeffding's inequality).** *Let $X_1, \cdots, X_N$ be independent bounded random variables such that $X_i \in [a_i, b_i]$, where $-\infty < a_i \leq b_i < \infty$. Then*

$$\mathbb{P}\left[\left|\frac{1}{N}\sum_{i=1}^{N}(X_i - \mathbb{E}X_i)\right| \geq t\right] \leq 2\exp\left[-\frac{2N^2t^2}{\sum_{i=1}^{N}(b_i - a_i)^2}\right].$$

To use Hoeffding's inequality, we need a bound $b$ for noise multipliers. If the noise distribution $\mathcal{D}$ has bounded support, this bound is natural. When the support of $\mathcal{D}$ is not bounded, we let

$$b = \frac{2}{c_0}\ln\left(c_0^{-1}2^{|A|+3}n\delta^{-1}\right).$$

When $n$ is sufficiently large,

$$\mathbb{P}[\xi \geq b] \leq \int_b^{\infty} \exp\left(-\frac{1}{2}c_0 x\right) dx \leq \frac{\delta}{2^{|A|+2}n}.$$

The probability that all $\xi(T)$ $(T \in \mathcal{L}_A)$ are bounded by $b$ is

$$\left(1 - \frac{\delta}{2^r n}\right)^{|\mathcal{L}_A|} \geq 1 - |\mathcal{L}_A| \cdot \frac{\delta}{2^{|A|+2}n} \geq 1 - \frac{\delta}{4}.$$

**Lemma D.7 (Difference between $\widetilde{\varphi_h}$ and $\widehat{\varphi_h}$).** *For any set $A \in \mathcal{I}(r-1)$, we assume $M \geq c^{\frac{1}{2}}\kappa^{-1} \cdot b(\ln(r-1)+2)\sqrt{n}$, where $c$ is the constant in Ineq. (9), and suppose that all $\xi(T)$ $(T \in \mathcal{L}_A)$ are bounded by $b$. Then*

$$\mathbb{P}\left[|\widehat{\varphi_h}(A) - \widetilde{\varphi_h}(A)| > \frac{\alpha}{2} \cdot \max_{S \in \mathcal{I}(r-1)} \varphi_h(S)\right] \leq \frac{\delta}{4}.$$

*Proof.* We apply Hoeffding's inequality by letting $X_j = \left(\sum_{T' \in \mathcal{L}_A} \tau_A(T')\right) \tilde{f}(T_j)$ and obtain

$$\mathbb{P}\left[|\widehat{\varphi_h}(A) - \widetilde{\varphi_h}(A)| \geq \frac{\alpha}{2} \cdot \max_{S \in \mathcal{I}(r-1)} \varphi_h(S)\right] \leq 2\exp\left[-\frac{M^2\varepsilon_0^2}{8b^2 \left(\sum_{T \in \mathcal{E}_A} \tau_A(T)\right)^2}\right].$$

By assumption that $M \geq c^{\frac{1}{2}}\kappa^{-1} \cdot b(\ln(r-1)+2)\sqrt{n}$ and $n \geq 192\kappa^2 c^{-1} \cdot \alpha^{-2}\ln(4\delta^{-1})$, the probability above is at most $\delta/4$. $\square$

For any $A \in \mathcal{I}(r-1)$, by taking a union bound, $|\widehat{\varphi_h}(A) - \varphi_h(A)| \leq \alpha \max_{S \in \mathcal{I}(r-1)} \varphi_h(S)$ fails with probability at most $\delta/2 + \delta/4 + (1-\delta/4)\delta/4 \leq \delta$. The lemma is proved.

### D.3 Proof of Lemma 4.5

We denote by $e^\star \in \arg\max_{e \in N \setminus S_L} \tilde{f}(S_L + e)$ the element added to $S_L$ at the $\tilde{f}$-maximization phase by Algorithm 2. We define two kinds of elements in $N \setminus S_L$. Say that an element $e$ is *good* if

$$f(S_L + e) \geq (1 - \frac{\varepsilon}{2})h(S_L), \tag{11}$$

and $e$ is *bad* if

$$f(S_L + e) < (1 - \varepsilon)h(S_L).$$

The set of good elements and that of bad ones are denoted as $\mathcal{G}$ and $\mathcal{B}$, respectively. Our goal is to prove that $e^\star$ is not bad with probability $1 - O\left(\frac{1}{\log n}\right)$.

First we use the following lemma to quantify the number of good elements.

**Lemma D.8 (The number of good elements).** *With probability $1 - \frac{1}{n}$, there are at least $\frac{\varepsilon n}{16H_{r-1}}$ good elements in $N \setminus S_L$, where $H_{r-1} = \sum_{i=1}^{r-1} \frac{1}{i}$.*

*Proof.* Suppose by contradiction that $|\mathcal{G}| < \frac{\varepsilon n}{16H_{r-1}}$. We will show that if there are only a few good elements in $N \setminus S_L$, a good element with very large $f$-value must exist. However, this contradicts how Algorithm 2 does local search, which should have changed some element in $S_L$ for the one with such large $f$-value.

By construction of function $h$ and definition of good elements, we have

$$h(S_L) = \frac{1}{n}\left(\sum_{e \in \mathcal{G}} f(S_L + e) + \sum_{e \in N \setminus \mathcal{G}} f(S_L + e)\right) \qquad \text{(Definition 4.2)}$$

$$< \frac{1}{n}\left(\sum_{e \in \mathcal{G}} f(S_L + e) + |N \setminus \mathcal{G}|\left(1 - \frac{\varepsilon}{2}\right)h(S_L)\right). \qquad \text{(Ineq. (11))}$$

Rewriting the inequality above gives

$$\sum_{e \in \mathcal{G}} f(S_L + e) > \left(n - \left(1 - \frac{\varepsilon}{2}\right)|N \setminus \mathcal{G}|\right) \cdot h(S_L) > \frac{\varepsilon n}{2} \cdot h(S_L),$$

where the last inequality holds since $|N \setminus \mathcal{G}| < n$. Hence, there must exist a good element $e_0$ such that

$$f(S_L + e_0) \geq \frac{\varepsilon n}{2|\mathcal{G}|} \cdot h(S_L) > 8H_{r-1} \cdot h(S_L). \tag{12}$$

Consider the set $S_L + e_0 - e_0'$, where $e_0'$ is an arbitrary element in $S_L$. On the one hand, we must have
$$\widehat{\varphi_h}(S_L + e_0 - e_0') \le (1 + \alpha)\widehat{\varphi_h}(S_L). \tag{13}$$
Otherwise, Algorithm 2 would have changed $e_0'$ for $e_0$ rather than returning $S_L$. We translate Ineq. (13) to that of $\varphi_h$. With probability $1 - O(\frac{1}{n})$,

$$
\begin{aligned}
\varphi_h(S_L + e_0 - e_0') &\le \widehat{\varphi_h}(S_L + e_0 - e_0') + \alpha \max_{S \in \mathcal{I}(r-1)} \varphi_h(S) && \text{(Lemma 4.4)} \\
&\le (1 + \alpha)\widehat{\varphi_h}(S_L) + \alpha \max_{S \in \mathcal{I}(r-1)} \varphi_h(S) && \text{(Ineq. (13))} \\
&\le (1 + \alpha)\left[\varphi_h(S_L) + 2\alpha \max_{S \in \mathcal{I}(r-1)} \varphi_h(S)\right] && \text{(Lemma 4.4)} \\
&\le (1 + \alpha)\left(1 + \frac{4\alpha}{1 - 2r\alpha}\right)\varphi_h(S_L) && \text{(Lemma C.1)} \\
&\le (1 + 10\alpha)\varphi_h(S_L). && (r\alpha < 1/4)
\end{aligned}
$$

On the other hand, for any $\alpha < \frac{1}{4}$ and $r \ge 2$,

$$
\begin{aligned}
\varphi_h(S_L + e_0 - e_0') &\ge h(S_L + e_0 - e_0') && \text{(Lemma C.5)} \\
&\ge f(S_L + e_0) - f(e_0') && \text{(monotonicity and submodularity of } f\text{)} \\
&\ge (8H_{r-1} - 1)h(S_L) && \text{(Ineq. (12)) and } f(e_0') \le f(S_L) \le h(S_L)) \\
&> (1 + 10\alpha)\frac{e}{e-1}H_{r-1}h(S_L) && (\alpha < 1/4 \text{ and } \frac{e}{e-1} < 2) \\
&\ge (1 + 10\alpha)\varphi_h(S_L). && \text{(Lemma C.5)}
\end{aligned}
$$

This constitutes a contradiction, and the lemma is proved. $\qquad\square$

Let $\xi_{\mathcal{G}}^* = \max_{e \in \mathcal{G}} \xi(S_L + e)$ and $\xi_{\mathcal{B}}^* = \max_{e \in \mathcal{B}} \xi(S_L + e)$ denote the largest noise multiplier on the sets belonging to $\{S_L + e : e \in \mathcal{G}\}$ and $\{S_L + e : e \in \mathcal{B}\}$, respectively. Next we show that with sufficient good elements guaranteed by Lemma D.8, $\xi_{\mathcal{G}}^*$ and $\xi_{\mathcal{B}}^*$ are close to each other with high probability.

If the distribution has bounded support and there is an atom at the supremum with some probability $p$, it is clear that both $\xi_{\mathcal{G}}^*$ and $\xi_{\mathcal{B}}^*$ reaches the supremum with high probability if $n$ is sufficiently large. Hence we focus on the case where the distribution $\mathcal{D}$ does not have bounded support in the following analysis. Recall that the probability density function $\rho(x)$ of the noise distribution $\mathcal{D}$ is $\exp(-\sum_i c_i x^{\gamma_i})$. We define two threshold $m_{\mathcal{G}}$ and $M_{\mathcal{B}}$:

$$m_{\mathcal{G}} = \left[\frac{1}{(1+\beta)c_0} \cdot \ln\left(\frac{|\mathcal{G}|}{\gamma_0 \ln n}\right)\right]^{\frac{1}{\gamma_0}}$$

and

$$M_{\mathcal{B}} = \left[\frac{1}{(1-\beta)c_0} \ln\left(\frac{|\mathcal{B}|\ln n}{\gamma_0}\right)\right]^{\frac{1}{\gamma_0}},$$

where we set $\beta = \left(\frac{1}{\ln n}\right)^{\frac{\gamma_0 - \gamma_1}{2\gamma_0}}$. Note that $m_{\mathcal{G}}$ and $M_{\mathcal{B}}$ are very close to each other in a sense that

$$\frac{m_{\mathcal{G}}}{M_{\mathcal{B}}} \ge \left[\frac{1-\beta}{1+\beta} \cdot \frac{\ln n - \ln(1/\varepsilon) - \ln\ln n - \ln(16\gamma_0 H_{r-1})}{\ln n + \ln\ln n - \ln\gamma_0}\right]^{\frac{1}{\gamma_0}} > 1 - \frac{\varepsilon}{2},$$

holds when $n \ge 6\exp\left(\left(\frac{1}{\varepsilon}\right)^{\frac{2\gamma_0}{\gamma_0 - \gamma_1}}\right)$. Our goal is to show that with high probability, $m_{\mathcal{G}}$ is a lower bound for $\xi_{\mathcal{G}}^*$, and $M_{\mathcal{B}}$ is an upper bound for $\xi_{\mathcal{B}}^*$. Before that, we need the following upper and lower bounds for the probability density function $\rho(x)$ of a noise distribution.

**Lemma D.9** (**Upper and lower bounds for $\rho(x)$ [20]**). *For any noise distribution $\mathcal{D}$ has a generalized exponential tail, there exists $n_0$ such that for any $n > n_0$ and $x \ge \left(\frac{2\sum_{i=1}|c_i|}{c_0}\right)^{\frac{1}{\gamma_0 - \gamma_1}} \cdot (\ln n)^{\frac{1}{2\gamma_0}}$, the probability density function $\rho(x) = \exp(-\sum_i c_i x^{\gamma_i})$ of $\mathcal{D}$ has the following upper and lower bound:*
$$(1+\beta)c_0 x^{\gamma_0 - 1}e^{-(1+\beta)c_0 x^{\gamma_0}} \le \rho(x) \le (1-\beta)c_0 x^{\gamma_0 - 1}e^{-(1-\beta)c_0 x^{\gamma_0}}.$$

**Lemma D.10 (Bounds for $\xi_{\mathcal{G}}^*$ and $\xi_{\mathcal{B}}^*$).** *With probability* $1 - O(\frac{1}{\log n})$, $\xi_{\mathcal{G}}^* \geq m_{\mathcal{G}}$ *and* $\xi_{\mathcal{B}}^* \leq M_{\mathcal{B}}$.

*Proof.* For a single sample $\xi$ drawn from noise distribution $\mathcal{D}$, we have

$$\mathbb{P}\left[\xi \leq m_{\mathcal{G}}\right] = 1 - \int_{m_{\mathcal{G}}}^{\infty} \rho(x)dx \leq 1 - \int_{m_{\mathcal{G}}}^{\infty} (1+\beta)c_0 x^{\gamma_0 - 1} e^{-(1+\beta)c_0 x^{\gamma_0}} dx = 1 - \frac{\ln n}{|\mathcal{G}|},$$

which follows from the lower bound shown in Lemma D.9 when $n$ is sufficiently large. The probability that all $\xi(S_L + e)$ $(e \in \mathcal{G})$ are bounded by $m_{\mathcal{G}}$ is at most

$$\left(1 - \frac{\ln n}{|\mathcal{G}|}\right)^{|\mathcal{G}|} \leq \frac{1}{e^{\ln n}} = \frac{1}{n}.$$

Hence $\xi_{\mathcal{G}}^* \geq m_{\mathcal{G}}$ holds with probability at least $1 - \frac{1}{n}$.

The probability that a single noise multiplier $\xi$ is bounded by $M_{\mathcal{B}}$ is at least

$$\mathbb{P}\left[\xi \leq M_{\mathcal{B}}\right] = 1 - \int_{M_{\mathcal{B}}}^{\infty} \rho(x)dx \geq 1 - \int_{M_{\mathcal{B}}}^{\infty} (1-\beta)c_0 x^{\gamma_0 - 1} e^{-(1-\beta)c_0 x^{\gamma_0}} dx = 1 - \frac{1}{|\mathcal{B}| \ln n},$$

which follows from the upper bound shown in Lemma D.9 when $n$ is sufficiently large. Then we obtain

$$\mathbb{P}[\xi_{\mathcal{B}}^* \leq M_{\mathcal{B}}] \geq \left(1 - \frac{1}{|\mathcal{B}| \ln n}\right)^{|\mathcal{B}|} \geq 1 - \frac{1}{\ln n}.$$

Therefore, by a union bound, $\xi_{\mathcal{G}}^* \geq m_{\mathcal{G}}$ and $\xi_{\mathcal{B}}^* \leq M_{\mathcal{B}}$ hold with probability $1 - O(\frac{1}{\log n})$. $\qquad\square$

With Lemma D.10, we have

$$\max_{e \in \mathcal{G}} \tilde{f}(S_L + e) \geq m_{\mathcal{G}} h(S_L) > \left(1 - \frac{\varepsilon}{2}\right) M_{\mathcal{B}} h(S_L) > \max_{e \in \mathcal{B}} \tilde{f}(S_L + e)$$

with probability $1 - O(\frac{1}{\log n})$. Thus a bad element will not be selected by Algorithm 2 at the $\tilde{f}$-maximization phase, i.e., $f(S_M) \geq (1-\varepsilon)h(S_L)$.

## E  Missing Proofs in Subsection 4.2

In this appendix, we provide the proofs that we omitted from Subsection 4.2.

### E.1  Proof of Lemma 4.8

For any set $S \subseteq N$ and $x \notin S$, by Definition 4.7, we have

$$h(S + x) - h(S) = \frac{1}{2^{|H|}} \sum_{H_j \subseteq H} \left[f(S \cup H_j + x) - f(S \cup H_j)\right],$$

which immediately indicates that $h$ is monotone. Moreover, for a set $T \subseteq S$,

$$h(T + x) - h(T) = \frac{1}{2^{|H|}} \sum_{H_j \subseteq H} \left[f(T \cup H_j + x) - f(T \cup H_j)\right]$$

$$\geq \frac{1}{2^{|H|}} \sum_{H_j \subseteq H} \left[f(S \cup H_j + x) - f(S \cup H_j)\right] \qquad \text{(submodularity of } f\text{)}$$

$$= h(S + x) - h(S).$$

Thus $h$ is also submodular. Finally, we lower bound $h_H(A)$ as below:

$$h_H(A) = \frac{1}{2^{|H|}} \sum_{H_j \subseteq H} f(A \cup H_j)$$

$$= \frac{1}{2^{|H|+1}} \sum_{H_j \subseteq H} [f(A \cup H_j) + f(A \cup H \setminus H_j)]$$

$$\geq \frac{1}{2^{|H|+1}} \sum_{H_j \subseteq H} [f(A \cup H) + f(A)] \qquad \text{(submodularity of } f)$$

$$= \frac{1}{2} [f(A \cup H) + f(A)].$$

## E.2  Proof of Lemma 4.9

For any set $A \subseteq N$, we rewrite $\varphi_{h_H}(A)$ as a linear combination of $f(T)$ according to Definitions 3.1 and 4.7:

$$\varphi_{h_H}(A) = \frac{1}{2^{|H|}} \sum_{T \subseteq A} \sum_{H_i \subseteq H} m_{|A|-1,|T|-1} f(T \cup H_i).$$

We will use Algorithm 6 to construct the oracle of approximate function $\widehat{\varphi_{h_H}}$. Similar to Lemma 4.4, we first show some bounds of coefficients $m_{|A|-1,|T|-1}$. With these bounds, we will then prove $\varphi_{h_H}$ and its noisy analogue $\widetilde{\varphi_{h_H}}$ are close to each other by Bernstein's inequality for sub-exponential variables. Finally, we use Hoeffding's inequality to show that with enough samples, $\widehat{\varphi_{h_H}}$ is close to $\widetilde{\varphi_{h_H}}$ and therefore a good approximation of $\varphi_{h_H}$.

---

**Algorithm 6:** Approximation of $\varphi_h$

---

**Input :** Noisy oracle $\tilde{f}$, a set $A \in \mathcal{I}_H(r)$ and $\varepsilon \in (0, 1/2)$

1 Let $M = 32 r H_r^2 \log^{\frac{5}{2}} n \cdot \varepsilon^{-1}$ and $s(A) = \sum_{T \subseteq A} m_{|A|-1,|T|-1}$

2 Construct a distribution $\nu_A(B)$ on set $2^N$ such that

$$\nu_A(B) = \begin{cases} \dfrac{m_{|A|-1,|T|-1}}{2^{|H|} s(A)} & \text{if there exists } T \subseteq A \text{ and } H_i \subseteq H \text{ such that } B = T \cup H_i \\ 0 & \text{otherwise} \end{cases}$$

3 Sample $M$ subsets $B_1, \cdots, B_M$ with distribution $\nu_A(B)$

4 $\widehat{\varphi_h}(A) \leftarrow s(A) \sum_{i=1}^{M} \tilde{f}(B_i)/M$

5 **return** $\widehat{\varphi_h}(A)$

---

**Lemma E.1 (Useful bounds of the coefficients).** *For any $A \subseteq N$ with size $|A| \geq 2$, we have*

$$(1) \quad \sum_{T \subseteq A} m_{|A|-1,|T|-1} \leq H_{|A|}; \qquad\qquad ([7, \text{Lemma 2}])$$

$$(2) \quad \sum_{T \subseteq A} m_{|A|-1,|T|-1}^2 \leq \frac{e^2}{(e-1)^2} \frac{2}{|A|} \leq 3.$$

*Proof.* It only remains to estimate the bound of $\sum_{T \subseteq A} m_{|A|-1,|T|-1}^2$. We have

$$\sum_{T \subseteq A} m_{|A|-1,|T|-1}^2 = \sum_{i=1}^{|A|} \left( \int_0^1 \frac{e^p}{e-1} p^{i-1}(1-p)^{|A|-i} dp \right)^2 \binom{|A|}{i} \qquad \text{(Definition 3.1)}$$

$$\leq \frac{e^2}{(e-1)^2} \sum_{i=1}^{|A|} \left( \int_0^1 p^{i-1}(1-p)^{|A|-i} dp \right)^2 \binom{|A|}{i} \qquad (p \in [0,1])$$

$$= \frac{e^2}{(e-1)^2} \sum_{i=1}^{|A|} \left( \frac{\Gamma(i)\Gamma(|A|-i+1)}{\Gamma(|A|+1)} \right)^2 \binom{|A|}{i} \qquad \text{(Ineq. (8))}$$

$$= \frac{e^2}{(e-1)^2} \left( \frac{1}{|A|} + \sum_{i=2}^{|A|} \frac{1}{i(|A|-i+1)} \frac{1}{\binom{|A|}{i-1}} \right) \qquad (\binom{|A|}{i-1} \geq |A|)$$

$$\leq \frac{e^2}{(e-1)^2} \frac{2}{|A|} \leq 3.$$

$\square$

To show that $\widehat{\varphi_{h_H}}$ is a $\left( \alpha = \frac{\varepsilon}{4r \ln r}, \delta = \frac{3}{n^6}, \mathcal{I}_H(r) \right)$-approximation of $\varphi_{h_H}$, it suffices to prove

$$\mathbb{P} \left[ |\widehat{\varphi_{h_H}}(A) - \varphi_{h_H}(A)| > \frac{\varepsilon}{4rH_r} \cdot \varphi_{h_H}(A) \right] \leq \frac{3}{n^6}.$$

First, we prove the the following lemma using concentration properties of sub-exponential random variables.

**Lemma E.2 (Difference between $\varphi_{h_H}$ and $\widetilde{\varphi_{h_H}}$).** *If $|H| = \lceil 3 \ln n \rceil$, we have*

$$\mathbb{P} \left[ |\widetilde{\varphi_{h_H}}(A) - \varphi_{h_H}(A)| > \frac{\varepsilon}{8rH_r} \varphi_{h_H}(A) \right] \leq \frac{2}{n^6}. \tag{14}$$

*Proof.* Let $\xi$ be a random variable with a generalized exponential tail distribution $\mathcal{D}$. The varaiable $\xi - 1$ has a finite sub-exponential norm $\kappa$ by Claim D.3. Let

$$X_{i,j} = \frac{1}{2^{|H|}} \cdot m_{|A|-1,|T_j|-1} f(T_j \cup H_i) \cdot (\xi_{T_j \cup H_i} - 1),$$

where $H_i \subseteq H$, $T_j \subseteq A$ and $i \in [2^{|H|}]$, $j \in [2^{|A|}]$. Applying Bernstein's inequality (Lemma D.4) gives

$$\mathbb{P} \left\{ |\widetilde{\varphi_{h_H}}(A) - \varphi_{h_H}(A)| > \frac{\varepsilon \varphi_{h_H}(A)}{8rH_r} \right\} \leq 2 \exp \left[ -c \min \left( \frac{\left( \frac{\varepsilon}{8rH_r} \right)^2 (\varphi_{h_H}(A))^2}{\sum_{i,j} \|X_{i,j}\|_{\psi_1}^2}, \frac{\frac{\varepsilon}{8rH_r} \varphi_{h_H}(A)}{\max_{i,j} \|X_{i,j}\|_{\psi_1}} \right) \right].$$

We can bound $\max_{i,j} \|X_{i,j}\|_{\psi_1}$ by

$$\max_{i,j} \|X_{i,j}\|_{\psi_1} \leq \frac{\kappa}{2^{|H|}} f(A \cup H) \cdot \max_{T_j} m_{|A|-1,|T_j|-1} \leq \frac{3}{2^{|H|}} \kappa \cdot f(A \cup H),$$

where we use the fact that for any $a, t \geq 0$,

$$m_{a,t} = \int_0^1 \frac{e^p}{e-1} p^t (1-p)^{a-t} dp \leq 3.$$

With Lemma E.1, $\sum_{i,j} \|X_{i,j}\|_{\psi_1}^2$ can be bounded by

$$\sum_{i,j} \|X_{i,j}\|_{\psi_1}^2 \leq \frac{1}{2^{|H|}} \sum_{T_j \subseteq A} m_{|A|-1,|T_j|-1}^2 \left( f(A \cup H) \right)^2 \kappa^2 \leq \frac{3}{2^{|H|}} \kappa^2 (f(A \cup H))^2.$$

For a sufficiently large $n \in \Omega(\varepsilon^{-4})$, we have $|H| \geq 3 \ln n \geq \ln \left( \frac{36864\kappa^2}{c} r^2 H_r^2 \ln n \cdot \varepsilon^{-2} \right)$. Then by Lemma 4.8 and direct calculations, we can conclude that

$$\min \left( \frac{\left( \frac{\varepsilon}{8rH_r} \right)^2 (\varphi_{h_H}(A))^2}{\sum_{i,j} \|X_{i,j}\|_{\psi_1}^2}, \frac{\frac{\varepsilon}{8rH_r} \varphi_{h_H}(A)}{\max_{i,j} \|X_{i,j}\|_{\psi_1}} \right) \geq \frac{6}{c} \ln n.$$

Therefore Ineq. (14) holds. $\square$

Next we prove the following lemma using Hoeffding's inequality (Lemma D.6).

**Lemma E.3 (Difference between $\widehat{\varphi_{h_H}}$ and $\widetilde{\varphi_{h_H}}$).** *If $n \in \Omega(\varepsilon^{-4})$, we have*

$$\mathbb{P}\left[|\widehat{\varphi_{h_H}}(A) - \widetilde{\varphi_{h_H}}(A)| > \frac{\varepsilon}{8rH_r}\varphi_{h_H}(A)\right] \leq \frac{1}{n^6}.$$

*Proof.* To use Hoeffding's inequality, we need a bound $b$ for noise multipliers. If the noise distribution $\mathcal{D}$ has bounded support, this bound is natural. When the support of $\mathcal{D}$ is not bounded, we let

$$b = \frac{16}{c_0}\ln\left(n \cdot c_0^{-\frac{1}{4}}\right).$$

Recall that we sample $M$ sets $B_1, \ldots, B_M$ to estimate $\widehat{\varphi_h}(A)$ in Algorithm 6. We say that event $\mathcal{E}_A$ happens if for all $i \in [M]$, we have $\xi_{B_i} \leq b$. If the support of $\mathcal{D}$ is bounded, event $\mathcal{E}_A$ trivially holds. Otherwise, for a distribution with unbounded support, recall that when

$$x \geq \max\left\{1, \left(\frac{2\sum_{i=1}|c_i|}{c_0}\right)^{\frac{1}{\gamma_0 - \gamma_1}}\right\},$$

we have the probability density $\rho(x) \leq e^{-\frac{1}{2}c_0 x^{\gamma_0}} \leq e^{-\frac{1}{2}c_0 x}$. When $n$ is sufficiently large, we have

$$\mathbb{P}[\xi \geq b] \leq \int_b^\infty \exp\left(-\frac{1}{2}c_0 x\right) dx \leq \frac{2}{c_0}\exp\left(-\frac{1}{2}c_0 b\right) = \frac{2}{n^8}.$$

Thus by taking a union bound, event $\mathcal{E}_A$ happens with probability at least $1 - 2M/n^8$.

Conditioned on event $\mathcal{E}_A$, consider the random variables $X_i = s(A)\tilde{f}(B_i)$. We bound $X_i$ by

$$\begin{aligned}
X_i &= s(A)\xi_{B_i}f(B_i) & \text{(Definition 2.4)} \\
&\leq H_r b f(B_i) & \text{(Lemma E.1 with $|A| \leq r$ and event $\mathcal{E}_A$)} \\
&\leq H_r b f(A \cup H). & \text{(monotonicity of $f$)}
\end{aligned}$$

Thus applying Hoeffding's inequality (Lemma D.6), we obtain

$$\mathbb{P}\left[|\widehat{\varphi_{h_H}}(A) - \widetilde{\varphi_{h_H}}(A)| > \frac{\varepsilon}{8rH_r}\varphi_{h_H}(A)\right] \leq \frac{2}{n^8},$$

which holds for a sufficiently large $n$ such that $\ln^2 n \geq b$. Therefore, $|\widehat{\varphi_h}(A) - \widetilde{\varphi_h}(A)| > \frac{\varepsilon}{8rH_r}\varphi_h(A)$ holds with probability at most

$$\frac{2M}{n^8} + \left(1 - \frac{2M}{n^8}\right)\frac{2}{n^8} = O(n^{-6}).$$

$\square$

Combining Lemma E.2 and E.3, we complete the proof of Lemma 4.9.

### E.3 Proof of Lemma 4.10

We first show that a random set $A$, which is uniformly selected from all subsets of $N$ with size $|H|$, satisfies that

$$\mathbb{E}[f(O^\star \setminus A)] \geq \left(1 - \frac{|H|}{r}\right)f(O^\star).$$

To prove this, we index $O^\star$ as $\{o_1, \cdots, o_r\}$ and denote by $I_A$ the indexes of elements in $O^\star \cap A$. Then we decompose $f(O^\star)$ as below:

$$f(O^\star) = \sum_{i=1}^r f(o_i \mid o_t, \ t \in [i-1]),$$

where $o_0$ is defined to be $\varnothing$. Similarly, we decompose $f(O^\star \setminus H)$ as

$$f(O^\star \setminus A) = \sum_{i \in [r] \setminus I_A} f(o_i \mid o_t, \ t \in [i-1] \text{ and } t \notin I_A)$$

$$\geq \sum_{i \in [r] \setminus I_A} f(o_i \mid o_t, \ t \in [i-1]). \qquad \text{(submodularity of } f)$$

Taking an expectation gives

$$\mathbb{E}\left[f(O^\star \setminus A)\right] \geq \mathbb{E}\left[\sum_{i \in [r] \setminus I_A} f(o_i \mid o_t, \ t \in [i-1])\right]$$

$$= \left(1 - \frac{|H|}{r}\right)\left(\sum_{i=1}^{r} f(o_i \mid o_t, \ t \in [i-1])\right) = \left(1 - \frac{|H|}{r}\right) f(O^\star).$$

Thus there must exist a set $A_0$ such that $f(O^\star \setminus A_0) \geq \left(1 - \frac{|H|}{r}\right) f(O^\star)$. Therefore, we have

$$\max_{S \in \mathcal{I}_H(r)} h_H(S) \geq h_H(O^\star \setminus A_0) \geq f(O^\star \setminus A_0) \geq \left(1 - \frac{|H|}{r}\right) f(O^\star).$$

# F  Proof of Theorem 5.1

In this appendix, we generalize Algorithm 2 to deal with general matroid constraints, which results in an algorithm (Algorithm 7) achieving the performance stated in Theorem 5.1.

## F.1  Useful notations and useful facts for Theorem 5.1

The local search procedure in this subsection is still based on the smoothing surrogate function $h$ in Definition 4.2. Similar to Lemma 4.4, we can construct a $(\alpha, \delta, \mathcal{I}(r-1) \cap \mathcal{I}(\mathcal{M}))$-approximation of auxiliary function $\varphi_h$.

**Lemma F.1 (Approximation of $\varphi_h$).** *Let $\alpha, \delta \in (0, 1/2)$ and assume $n \in \Omega(\alpha^{-2} \log(\delta^{-1}))$. There exists a value oracle $\mathcal{O}$ to an $(\alpha, \delta, \mathcal{I}(r-1) \cap \mathcal{I}(\mathcal{M}))$-approximation $\widehat{\varphi_h}$ of $\varphi_h$, which answering $\mathcal{O}(A)$ queries at most $O\left(\log r \cdot n^{\frac{1}{2}} \max\{r, \log n\}\right)$ times to $\tilde{f}$ for each set $A \in \mathcal{I}(r-1) \cap \mathcal{I}(\mathcal{M})$.*

Compared with the $(\alpha, \delta, \mathcal{I}(r-1))$-approximation in Lemma 4.4, an $(\alpha, \delta, \mathcal{I}(r-1) \cap \mathcal{I}(\mathcal{M}))$-approximation requires a smaller estimation error $\alpha \cdot \max\limits_{S \in \mathcal{I}(r-1) \cap \mathcal{I}(\mathcal{M})} \varphi_h(S)$ than $\alpha \cdot \max\limits_{S \in \mathcal{I}(r-1)} \varphi_h(S)$. However, the proof of Lemma 4.4 remains valid for Lemma F.1. This is because for any set $A \in \mathcal{I}(r-1) \cap \mathcal{I}(\mathcal{M})$, we can still bound $f(T)$ for all $T \in \mathcal{L}_A$ by $2 \cdot \max_{S \in \mathcal{I}(r-1) \cap \mathcal{I}(\mathcal{M})} \varphi_h(S)$ as in Eq. (10). Thus the concentration results (Lemma D.5 and D.7) in the proof hold as well.

Besides $\varphi_h$ (Definition 3.1) guiding the local search, we introduce another auxiliary function $\widehat{f_0}$ in this subsection, which will be used to compare the values of sets at the final step of Algorithm 7.

**Definition F.2 (Comparison auxiliary function).** For any set $S \subseteq N$, we define the comparison auxiliary function $f_0(S)$ as the expectation of $f(S - e)$ over a random element $e \in S$, i.e.,

$$f_0(S) = \frac{1}{|S|} \sum_{e \in S} f(S - e).$$

The comparison auxiliary function $f_0$ is close to $f$ in the sense that the following lemma holds.

**Lemma F.3 (Bounds of $f_0$).** *For any set $S \subseteq N$, we have*

$$\left(1 - \frac{1}{|S|}\right) f(S) \leq f_0(S) \leq f(S).$$

*Proof.* From monotonicity of $f$, we have $f_0(S) \leq f(S)$ immediately. We index the elements of $S$ as $s_i$, where $i \in [|S|]$. Then

$$f_0(S) = \frac{1}{|S|} \sum_{e \in S} [f(S) - f(e \mid S - e)] \qquad \text{(Definition F.2)}$$

$$= f(S) - \frac{1}{|S|} \sum_{e \in S} f(e \mid S - e)$$

$$\geq f(S) - \frac{1}{|S|} \sum_{i \in [|S|]} f(s_i \mid s_1, s_2, \cdots, s_{i-1}) \quad \text{(submodularity of } f\text{)}$$

$$= \left(1 - \frac{1}{|S|}\right) f(S). \qquad \qquad (\sum_{i \in [|S|]} f(s_i \mid s_1, \cdots, s_{i-1}) = f(S))$$

$\square$

Note that $f_0$ is also implicitly constructed since we only have a value oracle to $\tilde{f}$ instead of $f$. We denote $f_0$'s noisy analogue as $\widetilde{f_0}$, i.e.,

$$\widetilde{f_0}(S) = \frac{1}{|S|} \sum_{e \in S} \tilde{f}(S - e).$$

As $f_0(S)$ is based on an averaging set of size $|S|$, the following lemma indicates that $f_0$ and $\widetilde{f_0}$ are close to each other when $|S|$ is sufficiently large.

**Lemma F.4 (Concentration property of $f_0$).** *Let $\varepsilon, \delta \in (0, 1/2)$ and suppose that $|S| \geq \kappa \varepsilon^{-1} \ln(2\delta^{-1})$, where $\kappa$ is the sub-exponential norm of the noise multiplier. For any subset $S \subseteq N$, we have*

$$\mathbb{P}\left[\left|\widetilde{f_0}(S) - f_0(S)\right| > \varepsilon \cdot f_0(S)\right] \leq \delta.$$

*Proof.* For any set $T \in \{S - e : e \in S\}$, let $X_T = \xi(T) - 1$ and $a_T = \frac{1}{|S|} \cdot f(T)$. Recall that $X_T$ is a sub-exponential random variable with norm $\kappa$ by Claim D.3. Applying Bernstein's inequality for sub-exponential variables (Lemma D.4) gives

$$\mathbb{P}\left[\left|\widetilde{f_0}(S) - f_0(S)\right| > \varepsilon f_0(S)\right] \leq 2 \exp\left[-c \min\left(\frac{\varepsilon^2 |S|^2}{\kappa^2}, \frac{\varepsilon |S|}{\kappa}\right)\right].$$

By assumption that $|S| \geq \kappa \varepsilon^{-1} \ln(2\delta^{-1})$, the probability above is at most $\delta$. $\square$

### F.2 Algorithm for Theorem 5.1

We present Algorithm 7 that is a variant of Algorithm 2. The main difference is that Algorithm 7 contains a comparison phase (Line 4-7). Since $S_M$ may not be an independent set defined by $\mathcal{I}(\mathcal{M})$, Algorithm 7 compares $S_L$ returned by local search with the element $S_M \backslash S_L$ obtained at the $\tilde{f}$-maximization phase and returns the one with a larger $\widetilde{f_0}$ value.

---

**Algorithm 7:** Noisy local search subject to matroids with small ranks

---

**Input :** a value oracle to $\tilde{f}$, rank $r \in \Omega\left(\frac{1}{\varepsilon} \log\left(\frac{1}{\varepsilon}\right)\right) \cap O\left(n^{\frac{1}{3}}\right)$ and $\varepsilon \in (0, 1/2)$

1 Let $\widehat{\varphi_h}$ be a $\left(\frac{\varepsilon}{4r \ln r}, \frac{1}{(I+1)(r-1)n^2}, \mathcal{I}(r-1) \cap \mathcal{I}(\mathcal{M})\right)$-approximation of $\varphi_h$ as in Lemma F.1

2 $S_L \leftarrow \text{NLS}\left(\widehat{\varphi_h}, \mathcal{I}(r-1) \cap \mathcal{I}(\mathcal{M}), \Delta = \frac{\varepsilon}{4r \ln r}\right)$      ▷ Local search phase

3 $S_M \leftarrow S_L + \underset{e \in N \backslash S_L}{\arg\max} \tilde{f}(S_L + e)$      ▷ $\tilde{f}$-maximization phase

4 **if** $\widetilde{f_0}(S_L) \geq \frac{1}{2}\widetilde{f_0}(S_M)$ **then**

5    **return** $S_L$      ▷ Comparison phase

6 **else**

7    **return** $S_M \backslash S_L$

---

### F.3 Proof of Theorem 5.1

We first analyze the query complexity and then prove the approximation performance of Algorithm 7.

**Query complexity of Algorithm 7.** We call the noisy oracle $\tilde{f}$ only $(2r-1)$ times at the comparison phase. The query complexity of Algorithm 7 is dominated by the number of calls it makes at the local search phase and therefore of the same order $\tilde{O}(r^2 n^{\frac{3}{2}} \varepsilon^{-1})$ as Algorithm 2.

**Approximate ratio and success probability of Algorithm 7.** Recall that $O_f \in \arg\max\limits_{S \in \mathcal{I}(\mathcal{M})} f(S)$ denote an optimal solution to $f$. Similar to the analysis of Algorithm 2, it can be shown that we obtain a set $S_M$ at the end of the $\tilde{f}$-maximization phase such that with probability $1 - O(1/\log n)$,

$$f(S_M) \geq \left(1 - \frac{1}{e} - O(\varepsilon)\right) f(O_f). \tag{15}$$

Although $S_M$ may not be an independent set, we can decompose $S_M$ into two feasible parts $S_L \in \mathcal{I}(\mathcal{M})$ and $S_M \backslash S_L \in \mathcal{I}(\mathcal{M})$ and output one of them. The approximate ratio of the output set is guaranteed by the following lemma.

**Lemma F.5.** *We assume a sufficient small $\varepsilon$ and suppose that $r \in \Omega\left(\varepsilon^{-1}\log(\varepsilon^{-1})\right)$. Let $S_R$ denote the set returned by Algorithm 7. With probability at least $1 - 2\varepsilon^4$, we have*

$$f(S_R) \geq \left(\frac{1}{2} - O(\varepsilon)\right) f(S_M).$$

*Proof.* By Lemma F.4 with $\delta = \varepsilon^4$, we have

$$\left|\widetilde{f_0}(S) - f_0(S)\right| \leq \varepsilon f_0(S) \tag{16}$$

holds for both $S_L$ and $S_M$ with probability at least $1 - 2\varepsilon^4$. Suppose this is true.

If $\widetilde{f_0}(S_L) \geq \frac{1}{2}\widetilde{f_0}(S_M)$, then $S_R$ is $S_L$. We translate the inequality into that of $f_0$ by Ineq. (16):

$$f_0(S_L) \geq \frac{\widetilde{f_0}(S_L)}{1+\varepsilon} \geq \frac{\widetilde{f_0}(S_M)}{2(1+\varepsilon)} \geq \frac{1}{2} \cdot \frac{1-\varepsilon}{1+\varepsilon} \cdot f_0(S_M).$$

Then from Lemma F.3, we have

$$f(S_L) \geq f_0(S_L) \geq \frac{1}{2} \cdot \frac{1-\varepsilon}{1+\varepsilon} \cdot f_0(S_M) \geq \frac{1}{2} \cdot \frac{1-\varepsilon}{1+\varepsilon} \left(1 - \frac{1}{r}\right) f(S_M).$$

For any $r > 1/\varepsilon$, the inequality above implies that

$$f(S_L) \geq \left(\frac{1}{2} - 2\varepsilon\right) f(S_M).$$

Algorithm 7 returns $S_M \setminus S_L$ if $\widetilde{f_0}(S_L) < \frac{1}{2}\widetilde{f_0}(S_M)$. Similarly, we convert this by Ineq. (16) into the following inequality:

$$f_0(S_L) \leq \frac{\widetilde{f_0}(S_L)}{1-\varepsilon} \leq \frac{1}{2} \cdot \frac{\widetilde{f_0}(S_M)}{1-\varepsilon} \leq \frac{1}{2} \cdot \frac{1+\varepsilon}{1-\varepsilon} f_0(S_M).$$

By Lemma F.3,

$$f(S_L) \leq \frac{r-1}{r-2} f_0(S_L) < \frac{1}{2} \cdot \frac{1+\varepsilon}{1-\varepsilon} \cdot \frac{r-1}{r-2} f_0(S_M) \leq \frac{1}{2} \cdot \frac{1+\varepsilon}{1-\varepsilon} \cdot \frac{r-1}{r-2} f(S_M),$$

which is directly followed by

$$f(S_L) < \frac{1}{2}(1 + 5\varepsilon)f(S_M)$$

if $\varepsilon < 1/3$ and $r \geq 2/\varepsilon$. Since submodularity of $f$ indicates that $f(S_L) + f(S_M \setminus S_L) \geq f(S_M)$, we have

$$f(S_M \setminus S_L) \geq \left(\frac{1}{2} - 3\varepsilon\right) f(S_M).$$

$\square$

Combining Ineq. (15) and Lemma F.5, we prove that Algorithm 7 achieves a $((1 - 1/e)/2 - O(\varepsilon))$-approximation with probability $1 - O(\varepsilon^4)$.

# G  Proof of Theorem 5.2

This appendix describes a generalization (Algorithm 8) of Algorithm 3, whose performance matches Theorem 5.2.

## G.1  Useful facts for Theorem 5.2

Our algorithm and analysis in this subsection are based on the smoothing surrogate function $h_H$ in Definition 4.7 and comparison auxiliary function $f_0$ in Definition F.2. We introduce a contraction matroid $\mathcal{I}_H(\mathcal{M})$ confined on $N \backslash H$. Any feasible set $S \in \mathcal{I}_H(\mathcal{M})$ can be converted to an independent set subject to $\mathcal{I}(\mathcal{M})$ by adding $H$.

**Definition G.1 (Contraction matroid).** For any subset $H$ and matroid constraint $\mathcal{I}(\mathcal{M})$, we define contraction matroid over $N \setminus H$ as the following:

$$\mathcal{I}_H(\mathcal{M}) = \{S \subseteq N \setminus H : S \cup H \in \mathcal{I}(\mathcal{M})\}.$$

Similar to Lemma 4.9, the following lemma indicates that the auxiliary function $\varphi_{h_H}$ can be well approximated.

**Lemma G.2 (Approximation of $\varphi_{h_H}$).** Let $\varepsilon > 0$ and for sufficiently large $n \in \Omega(\frac{1}{\varepsilon^4})$. If $|H| \geq 3 \ln n$, there exists a value oracle $\mathcal{O}$ to an $(\frac{\varepsilon}{4r \ln r}, \frac{3}{n^6}, \mathcal{I}_H(\mathcal{M}))$-approximation $\widehat{\varphi_{h_H}}$ of $\varphi_{h_H}$, in which answering $\mathcal{O}(A)$ queries at most $\tilde{f}$'s oracle $O(r\varepsilon^{-1} \log^{\frac{5}{2}} n \log^2 r)$ times of $\tilde{f}$ for each set $A \in \mathcal{I}$.

The proof of Lemma 4.9 actually demonstrates a stricter claim that for any set $A \in \mathcal{I}_H(r)$, $\mathbb{P}\left[|\widehat{\varphi_h}(A) - \varphi_h(A)| > \frac{\varepsilon}{4r \ln r} \varphi_{h_H}(A)\right] \leq \frac{3}{n^6}$. Since $\mathcal{I}_H(\mathcal{M}) \subseteq \mathcal{I}_H(r)$, the proof remains valid for Lemma G.2.

## G.2  Algorithm for Theorem 5.2

---

**Algorithm 8:** Noisy local search subject to matroids with large ranks

---

**Input :** a value oracle to $\tilde{f}$, rank $r \in \Omega\left(n^{\frac{1}{3}}\right)$, and $\varepsilon \in (0, 1/2)$

1  Arbitrarily select a basis $B_0 \subseteq N$ and spilt $B_0$ into two parts $H_1, H_2$ with size $\lfloor \frac{r}{2} \rfloor$ or $\lfloor \frac{r}{2} \rfloor + 1$
2  **for** $t = 1, 2$ **do**
3  $\quad$ Let $\widehat{\varphi_{h_{H_t}}}$ be a $(\alpha = \frac{\varepsilon}{4r \ln r}, \delta = \frac{3}{n^6}, \mathcal{I}_{H_t}(\mathcal{M}))$-approximation of $\varphi_{h_{H_t}}$ as in Lemma G.2
4  $\quad$ $S_t \leftarrow \text{NLS}\left(\widehat{\varphi_{h_{H_t}}}, \mathcal{I}_{H_t}(\mathcal{M}), \Delta = \frac{\varepsilon}{4r \ln r}\right)$  $\qquad \triangleright$ `Local search phase`
5  Let $i^\star \leftarrow \underset{t \in \{1,2\}}{\arg\max} \ \tilde{f_0}(S_t \cup H_t)$  $\qquad\qquad\qquad \triangleright$ `Comparison phase`
6  **return** $S_{i^\star} \cup H_{i^\star}$

---

We present Algorithm 8 that contains two phases: a local search phase (Lines 2-4) and a comparison phase (Line 5). Algorithm 8 arbitrarily splits a basis (maximal independent set) into two parts of almost the same size, and it then grows each one of them into a basis using Algorithm 1, respectively. Finally the algorithm compares two solutions $S_t \cup H_t$ ($t = 1, 2$) and outputs the better one in terms of $\tilde{f_0}$.

## G.3  Proof of Theorem 5.2

To prove Theorem 5.2, we analyze the query complexity and approximation performance of Algorithm 7 as below.

**Query Complexity of Algorithm 8.**  The number of calls to $\tilde{f}$ during local search phase dominates the query complexity of Algorithm 8 as it queries only $2r$ times at the comparison phase. Algorithm 8 runs the local search procedure twice, and thus the query complexity of Algorithm 8 is of the same order $O(n^6 \varepsilon^{-1})$ as that of Algorithm 3.

**Approximation performance analysis of Algorithm 8.** Similar to the analysis of Algorithm 3, for $t \in \{1, 2\}$, Corollary 3.4 implies that with probability $1 - O\left(\frac{1}{n^2}\right)$,

$$f(S_t \cup H_t) \geq \left(1 - \frac{1}{e} - \varepsilon\right) \max_{S \subseteq \mathcal{I}_{H_t}(\mathcal{M})} h_{H_t}(S). \tag{17}$$

Recall that $O^\star \in \arg\max_{S \in \mathcal{I}(\mathcal{M})} f(S)$ denotes an optimal solution. The following lemma relates $\max_{S \subseteq \mathcal{I}_{H_t}(\mathcal{M})} h_{H_t}(S_t)$ to $f(O^\star)$.

**Lemma G.3.** $f(O^\star) \leq \max_{S \subseteq \mathcal{I}_{H_1}(\mathcal{M})} h_{H_1}(S_1) + \max_{S \subseteq \mathcal{I}_{H_2}(\mathcal{M})} h_{H_2}(S_2).$

*Proof.* We first introduce a lemma concerning the structure of matroids.

**Lemma G.4** (See [10]). *Given two bases $B_1$, $B_2$ of a matroid $\mathcal{M}$ a partition $B_1 = X_1 \cup Y_1$, there is a partition $B_2 = X_2 \cup Y_2$ such that $X_1 \cup Y_2$ and $X_2 \cup Y_1$ are both bases of $\mathcal{M}$.*

The lemma above indicates that there is a partition of $O^\star = O_1 \cup O_2$ such that $O_t \cup H_t \in \mathcal{I}(\mathcal{M})$ for $t \in \{1, 2\}$. Hence we have

$$
\begin{aligned}
\max_{S \subseteq \mathcal{I}_{H_1}(\mathcal{M})} h_1(S_1) + \max_{S \subseteq \mathcal{I}_{H_2}(\mathcal{M})} h_2(S_2) \; &\geq \; h_{H_1}(O_1) + h_{H_2}(O_2) \\
&\geq \; f(O_1) + f(O_2) && \text{(monotonicity of } f\text{)} \\
&\geq \; f(O_1 \cup O_2) = f(O^\star). && \text{(submodularity of } f\text{)}
\end{aligned}
$$

$\square$

Finally, we show that comparison with noisy auxiliary function $\widetilde{f}_0$ causes only a small loss in the approximate ratio.

**Claim G.5.** *We assume a sufficient small $\varepsilon$ and suppose that $r \in \Omega\left(n^{\frac{1}{3}}\right)$. Let $S_R$ denote the set returned by Algorithm 8. With probability at least $1 - \frac{2}{n^4}$, we have*

$$f(S_R) \geq \left(\frac{1}{2} - O(\varepsilon)\right)(f(S_1 \cup H_1) + f(S_2 \cup H_2)).$$

*Proof.* By Lemma F.4 with $\delta = 1/n^4$, we have

$$\left|\widetilde{f}_0(S) - f_0(S)\right| \leq \varepsilon f_0(S) \tag{18}$$

holds for both $S_1 \cup H_1$ and $S_2 \cup H_2$ with probability at least $1 - 2/n^4$. Suppose this is true. Then for $t \in \{1, 2\}$, we have

$$
\begin{aligned}
f_0(S_R) &\geq \frac{1}{1+\varepsilon} \cdot \widetilde{f}_0(S_R) && \text{(Ineq. (18))} \\
&\geq \frac{1}{2(1+\varepsilon)} \cdot \left(\widetilde{f}_0(S_1 \cup H_1) + \widetilde{f}_0(S_2 \cup H_2)\right) && (\widetilde{f}_0(S_R) = \max_{t \in \{1,2\}} \widetilde{f}_0(S_t \cup H_t)) \\
&\geq \frac{1}{2} \cdot \frac{1-\varepsilon}{1+\varepsilon} \cdot (f_0(S_1 \cup H_1) + f_0(S_2 \cup H_2)). && \text{(Ineq. (18))} \tag{19}
\end{aligned}
$$

We convert this inequality of $f_0$ to that of $f$:

$$
\begin{aligned}
f(S_R) &\geq f_0(S_R) && \text{(Lemma F.3)} \\
&\geq \frac{1}{2} \cdot \frac{1-\varepsilon}{1+\varepsilon} \cdot (f_0(S_1 \cup H_1) + f_0(S_2 \cup H_2)) && \text{(Ineq. (19))} \\
&\geq \frac{1}{2} \cdot \frac{1-\varepsilon}{1+\varepsilon} \cdot \left(1 - \frac{1}{r}\right)(f(S_1 \cup H_1) + f(S_2 \cup H_2)) && \text{(Lemma F.3)} \\
&\geq \left(\frac{1}{2} - 2\varepsilon\right)(f(S_1 \cup H_1) + f(S_2 \cup H_2)), && (\frac{1-\varepsilon}{1+\varepsilon} \geq 1 - 2\varepsilon \text{ and } r \geq 1/\varepsilon)
\end{aligned}
$$

which completes the proof. $\square$

Combining Ineq. (17), Claim G.3 and Claim G.5, we can conclude that with probability $1 - O\left(\frac{1}{n^2}\right)$,

$$f(S_{i^\star} \cup H_{i^\star}) \geq \left(\frac{1}{2}\left(1 - \frac{1}{e}\right) - O(\varepsilon)\right) f(O^\star),$$

which matches Theorem 5.2.

## H  Approximation algorithm for strongly base-orderable matroid constraints

In this section, we consider a special family of matroids, called *strongly base-orderable matroids*.

**Definition H.1 (Strongly base-orderable matroid).** A matroid $\mathcal{M}$ is strongly base-orderable if given any two bases $B_1$ and $B_2$, there is a bijection $\sigma : B_1 \to B_2$ such that for any $X \subseteq B_1$, $(B_1 \setminus X) \cup \sigma(X)$ is a basis, and $(B_2 \setminus \sigma(X)) \cup X$ is a basis.

As is evident from Definition H.1, the cardinality constraint which we discuss in Section 4 is a special case of strongly base-orderable matroid. Moreover, this family of matroids includes many typical matroids as well, such as partition matroids (Definition A.3) and transversal matroids.

Next we present an algorithm (Algorithm 9) and its analysis (Theorem H.2) that achieves near-tight approximation guarantees subject to strongly base-orderable matroids with rank $r \in \Omega\left(n^{\frac{1}{3}}\right)$.

**Theorem H.2 (Algorithmic results for cardinality constraints when $r \in \Omega\left(n^{\frac{1}{3}}\right)$).** *Let $\varepsilon > 0$ and assume $n \in \Omega\left(\frac{1}{\varepsilon^4}\right)$ is sufficiently large. For any $r \in \Omega\left(n^{\frac{1}{3}}\right)$, there exists an algorithm that returns a $\left(1 - \frac{1}{e} - O(\varepsilon)\right)$-approximation for Problem 2.6 under a strongly base-orderable matroid constraint $\mathcal{I}(\mathcal{M})$, with probability at least $1 - O\left(\frac{1}{n}\right)$ and query complexity at most $O(n^7 \varepsilon^{-1})$ to $\tilde{f}$.*

Compared with Theorem 5.2, Theorem H.2 improves the approximate ratio to $\left(1 - \frac{1}{e} - O(\varepsilon)\right)$ for the strongly base-orderable matroids, which have stronger exchangeable structures.

### H.1  Algorithm for Theorem H.2

Similar to Algorithm 8, Algorithm 9 also contains two phases: a local search phase (Lines 3-5) and a comparison phase (Line 6). The local search procedure is based on the smoothing surrogate function $h_H$ in Definition 4.7, while the comparison auxiliary function $f_0$ in Definition F.2 is used at Line 6. Algorithm 8 differs from Algorithm 9 by running the local search procedure (Algorithm 1) $\lfloor \frac{r}{l} \rfloor$ times rather than twice, with different smoothing surrogate functions $h_{H_t}$.

---

**Algorithm 9:** Noisy local search under strongly base-orderable matroid constraints

---

**Input :** a value oracle to $\tilde{f}$, rank $r \in \Omega\left(n^{\frac{1}{3}}\right)$, and $\varepsilon \in (0, 1/2)$

1 Let $l \leftarrow \lceil 3 \ln n \rceil$
2 Arbitrarily select a basis $B_0$ and arbitrarily spilt $B_0$ into $\lfloor \frac{r}{l} \rfloor$ parts $H_1, \cdots, H_{\lfloor \frac{r}{l} \rfloor}$ with size $l$ or $l + 1$
3 **for** $t = 1, \cdots, \lfloor \frac{r}{l} \rfloor$ **do**
4 $\quad$ Let $\widehat{\varphi_{h_{H_t}}}$ be a $(\alpha = \frac{\varepsilon}{4r \ln r}, \delta = \frac{3}{n^6}, \mathcal{I}_{H_t}(\mathcal{M}))$-approximation of $\varphi_{h_{H_t}}$ as in Lemma G.2
5 $\quad$ $S_t \leftarrow \text{NLS}\left(\widehat{\varphi_{h_{H_t}}}, \mathcal{I}_{H_t}(\mathcal{M}), \Delta = \frac{\varepsilon}{4r \ln r}\right)$ $\qquad\qquad$ ▷ Local search phase
6 Let $i^\star \leftarrow \arg\max_{t \in [\lfloor \frac{r}{k} \rfloor]} \widetilde{f}_0(S_t \cup H_t)$ $\qquad\qquad$ ▷ Comparison phase
7 **return** $S_{i^\star} \cup H_{i^\star}$

---

### H.2  Proof of Theorem H.2

We analyze the query complexity and approximation performance of Algorithm 9 to prove Theorem H.2. To simplify notation, we use $h_t$ to stand for $h_{H_t}$, $\widehat{\varphi_{h_t}}$ for $\widehat{\varphi_{h_{H_t}}}$, and $\mathcal{I}_t(\mathcal{M})$ for $\mathcal{I}_{H_t}(r)$ in the analysis.

**Query Complexity of Algorithm 9.** Similar to the proof of Theorem 5.2, Algorithm 9 runs the local search procedure $\lfloor r/l \rfloor$ times, and thus the query complexity of Algorithm 9 is at most $O(n^7 \varepsilon^{-1})$.

**Approximation performance analysis of Algorithm 9.** Similarly, from mononicity of $f$ and Corollary 3.4, we have that

$$f(S_t \cup H_t) \geq \left(1 - \frac{1}{e} - \varepsilon\right) \max_{S \subseteq \mathcal{I}_t(\mathcal{M})} h_t(S) \tag{20}$$

holds for all $t \in [\lfloor r/l \rfloor]$ with probability $1 - O\left(\frac{r}{n^2}\right)$. Let $O^\star \in \arg\max_{S \in \mathcal{I}(\mathcal{M})} f(S)$ be an optimal solution. The following lemma relates $\max_{S \subseteq \mathcal{I}_t(\mathcal{M})} h_t(S)$ to $f(O^\star)$.

**Lemma H.3.** *We have*

$$\mathbb{E}_{t \sim \mathcal{U}} \left[ \max_{S \subseteq \mathcal{I}_t(\mathcal{M})} h_t(S) \right] \geq \left(1 - \frac{l+1}{r}\right) f(O^\star),$$

*where $\mathcal{U}$ is a uniform distribution over $[\lfloor \frac{r}{l} \rfloor]$.*

*Proof.* By Definition H.1, there is a bijection $\sigma : B_0 \to O^\star$ between the elements in $B_0$ and $O^\star$ such that for all $t \in [\lfloor \frac{r}{l} \rfloor]$, $(O^\star \setminus \sigma(H_t)) \in \mathcal{I}_t(\mathcal{M})$. The following lemma gives a lower bound of $f(O^* \setminus \sigma(H_t))$ in expectation.

**Lemma H.4.** *Given an arbitrary partition $O_1, \cdots, O_{\lfloor r/l \rfloor}$ of $O^\star$ such that $|O_t| \in \{l, l+1\}$ ($t \in [\lfloor \frac{r}{l} \rfloor]$), we have*

$$\mathbb{E}_{t \sim \mathcal{U}} [f(O^\star \setminus O_t)] \geq \left(1 - \frac{l+1}{r}\right) f(O^\star),$$

*where $\mathcal{U}$ is a uniform distribution over $[\lfloor \frac{r}{l} \rfloor]$.*

*Proof.* We index $O^\star$ as $\{o_1, \cdots, o_r\}$ and denote by $I_t$ the indexes of the elements in $O_t$ for $t \in [\lfloor \frac{r}{l} \rfloor]$. We can decompose $f(O^\star)$ as

$$f(O^\star) = \sum_{i=1}^{r} f(o_i \mid o_k, k \in [i-1]).$$

$f(O^\star \setminus O_t)$ can also be decomposed as below:

$$f(O^\star \setminus O_t) = \sum_{i \in [r] \setminus I_t} f(o_i \mid o_k, k \in [i-1] \text{ and } k \notin I_t)$$

$$\geq \sum_{i \in [r] \setminus I_t} f(o_i \mid o_k, k \in [i-1]). \qquad \text{(submodularity of } f)$$

Taking an expectation over $t \in [\lfloor \frac{r}{l} \rfloor]$ gives

$$\mathbb{E}_{t \sim \mathcal{U}} [f(O^\star \setminus O_t)] \geq \mathbb{E}_{t \sim \mathcal{U}} \left[ \sum_{i \in [r] \setminus I_t} f(o_i \mid o_k, k \in [i-1]) \right]$$

$$= \left(1 - \frac{1}{\lfloor \frac{r}{l} \rfloor}\right) \sum_{i=1}^{r} f(o_i \mid o_k, k \in [i-1]) \geq \left(1 - \frac{l+1}{r}\right) f(O^\star).$$

$\square$

With Lemma H.4, we can prove Lemma H.3 since

$$\mathbb{E}_{t \sim \mathcal{U}} \left[ \max_{S \subseteq \mathcal{I}_t(\mathcal{M})} h_t(S) \right] \geq \mathbb{E}_{t \sim \mathcal{U}} [h_t(O^\star \setminus \sigma(H_t))] \qquad ((O^\star \setminus \sigma(H_t)) \in \mathcal{I}_t(\mathcal{M}))$$

$$\geq \underset{t\sim\mathcal{U}}{\mathbb{E}} \left[ f(O^\star \setminus \sigma(H_t)) \right] \qquad \text{(submodularity of } f\text{)}$$

$$\geq \left( 1 - \frac{l+1}{r} \right) f(O^\star). \qquad \text{(Lemma H.4)}$$

$\square$

Finally, similar to Lemma G.5, the following lemma shows that comparison with $\widetilde{f}_0$ causes only a small loss in the approximate ratio.

**Lemma H.5.** *We assume a sufficient small $\varepsilon$ and suppose that $r \in \Omega\left(n^{\frac{1}{3}}\right)$. With probability at least $1 - \frac{2r}{n^4}$, we have*

$$f(S_{i^\star} \cup H_{i^\star}) \geq (1 - O(\varepsilon)) \underset{t\sim\mathcal{U}}{\mathbb{E}} \left[ f(S_t \cup H_t) \right].$$

The proof idea of this lemma is the same as that of Lemma G.5. The only difference is that when proving Lemma H.5, we need to take a union bound of probability that $\widetilde{f}_0(S_t \cup H_t)$ is close to $f(S_t \cup H_t)$ for all $i \in [\lfloor r/l \rfloor]$, thus the success probability in Lemma H.5 is at least $1 - \frac{2r}{n^4}$.

Now we arrive that with probability $1 - O\left(\frac{1}{n}\right)$, we have

$$\begin{aligned}
f(S_{i^\star} \cup H_{i^\star}) &\geq (1 - O(\varepsilon)) \underset{t\sim\mathcal{U}}{\mathbb{E}} \left[ f(S_t \cup H_t) \right] & \text{(Lemma H.5)} \\
&\geq \left( 1 - \frac{1}{e} - O(\varepsilon) \right) \underset{t\sim\mathcal{U}}{\mathbb{E}} \left[ \max_{S \subseteq \mathcal{I}_t(\mathcal{M})} h_t(S) \right] & \text{(Ineq. (20))} \\
&\geq \left( 1 - \frac{1}{e} - O(\varepsilon) \right) f(O^\star), & \text{(Lemma H.3 )}
\end{aligned}$$

which matches the approximation performance of Algorithm 9 in Theorem H.2.