# OpenReview forum: "Efficient Submodular Optimization under Noise: Local Search is Robust"
_NeurIPS.cc/2022/Conference — NeurIPS 2022 Accept_

### Official Review · Reviewer_6S7j · 2022-07-02

**Rating:** 7
**Confidence:** 3
**Soundness:** 3 good
**Presentation:** 4 excellent
**Contribution:** 3 good

**Summary:**

The paper focuses on the problem of monotone submodular maximization subject to matroid constraints, when the function is only accessible through a noisy oracle. In particular, they provide an algorithm that with high probability outputs a solution with near-optimal (1 - 1/e - O(\epsilon)-approximation guarantee. Notably, they deliver this result with a polynomial computational complexity in 1/\epsilon, while the current works in the literature have an exponential complexity.

**Questions:**

The claims that presented earlier in the paper in a less formal manner can be a bit misleading and undermine the main results of the paper. For example:
(a) In the abstract, it is stated that the algorithm provides a 1 - 1/e - O(\epsilon) guarantee without introducing \epsilon. It is my understanding that this \epsilon is an arbitrary value (given Theorems 5.1 and 5.2). If so, could the authors please clear this up in the abstract? As otherwise the provided bound can be quite far from the optimal.

(b) In Theorem 1.1, the success probability of the algorithm is stated to be 1 - O(1). However, given Theorems 5.1 and 5.2, it seems that this is not correct as the probability either depends on n or \epsilon and is not a constant.

**Ethics Review Area:**

["I don’t know"]

**Limitations:**

Yes.

**Strengths And Weaknesses:**

Strengths:
(a) The paper studies an interesting generic problem with broad applications in the data mining and machine learning areas.
(b) The paper is very well-written and easy to follow. In particular, the authors make the transition into their main results smooth by providing sufficient intuitive explanations and motivations. Moreover, they justly credit the related the works and clearly distinguish their contributions from the existing methods or definitions.
(c) They provide strong results and considerable improvements over the state-of-the-art.

Weaknesses:
Overall I find the results quite strong. I just have few minor comments on the presentation of the results, which I state under the Questions.

---

> ### Author Response · Authors · 2022-08-02
> **Re: Official Review of Paper4250 by Reviewer 6S7j**
>
> Thank you for your positive review and feedback. We answer your specific questions below.
>
> > ''In the abstract, it is stated that the algorithm provides a $1 - 1/e - O(\varepsilon)$ guarantee without introducing $\varepsilon$. It is my understanding that this $\varepsilon$ is an arbitrary value (given Theorems 5.1 and 5.2). If so, could the authors please clear this up in the abstract? As otherwise the provided bound can be quite far from the optimal.''
>
> Yes, $\varepsilon$ is an arbitrarily small value. We have clarified it in the revised pdf.
>
> > ''In Theorem 1.1, the success probability of the algorithm is stated to be $1 - O(1)$. However, given Theorems 5.1 and 5.2, it seems that this is not correct as the probability either depends on n or $\varepsilon$ and is not a constant.''
>
> The success probability of the algorithm is $1-o(1)$ instead of $1-O(1)$, where $o(1)$ is a small quantity depending on $n$ or $\varepsilon$.

---

### Official Review · Reviewer_ECg7 · 2022-07-11

**Rating:** 7
**Confidence:** 3
**Soundness:** 4 excellent
**Presentation:** 3 good
**Contribution:** 3 good

**Summary:**

This paper studies the problem of maximizing a submodular set function when function values are given by a noisy oracle. The authors propose 2 novel auxiliary functions which can be queried efficiently in order to improve the algorithms' candidate solutions via local search. The approach for cardinality constraints is then extended to work with matroid constraints.

**Questions:**

- Can the surrogates be used in the noise-free setting to improve the query complexity of local search? Does this improve on the guarantees in reference [5] or simply generalize them when the oracle has noise?
- Is there a clear mapping from noise parameters $c_i$, $\gamma_i$ to auxiliary function constants $\alpha$, $\delta$?

[5] Yuval Filmus and Justin Ward. Monotone submodular maximization over a matroid via non-oblivious local search. SIAM Journal on Computing, 43(2):514–542, 2014.

**Limitations:**

This paper describes limitations as opportunities for future work, but does not address any potential negative social impact.

**Strengths And Weaknesses:**

## Originality/ Significance
- Interesting theoretical contributions which lead to more practical ways to handle noisy function evaluation
- Cites previous related work where appropriate

## Quality
- Smoothing surrogate auxiliary functions appear to be a novel, elegant approach that leads to exponential improvement in query complexity compared to previous work

## Clarity
- This paper is generally well organized. However I have 2 small suggestions:
    - State the note after Theorem 3.3 as a separate corollary, showing intermediate steps in the Appendix. this is important because subsequent theorems appear to include an extra $\log r$ term
    - The second paragraph of related work on noise-free optimization with more complex constraints is not relevant to the paper. This could be moved to the Appendix or removed altogether
- "approximate ratio" is used throughout the paper where "approximation ratio" is more standard

---

> ### Author Response · Authors · 2022-08-02
> **Re: Official Review of Paper4250 by Reviewer ECg7**
>
> We would like to thank the reviewer for their positive comments and valuable suggestions. We have improved the clarity of our paper in the revised pdf as suggested.
>
> > ''Can the surrogates be used in the noise-free setting to improve the query complexity of local search? Does this improve on the guarantees in reference [5] or simply generalize them when the oracle has noise?''
>
> The surrogates cannot be used to improve the query complexity of local search in the absence of noise. Compared to $\varphi_f(S)$ for the noise-free setting, the auxiliary functions $\varphi_h(S)$ (by introducing $h$ in Definitions 4.2 and 4.7) can be written as a linear combination of a **larger** collection of $f(T)$s. Due to this fact, the estimation error of $\varphi_h(S)$ can not be smaller than that of $\varphi_f(S)$ by applying Hoeffding's inequality (Lemma D.6).
>
> Our framework is a generalization of [5] in the presence of noise.
>
> > ''Is there a clear mapping from noise parameters $c_i, \gamma_i$ to auxiliary function constants $\alpha$, $\delta$?''
>
> The sub-exponential norm $\kappa$ (Definition D.2) of the noise distribution depends on the parameters $c_i, \gamma_i$ of the density function. A small $\kappa$ indicates a concentrated noise distribution. As mentioned in Lemma D.5, the relationship among $\alpha$, $\delta$ and $\kappa$ can be expressed as $\alpha^{-2}\ln(4\delta^{-1}) \leq c\cdot n\kappa^{-2}$, where $c$ is an absolute constant. Generally speaking, the more concentrated the noise is ($\kappa \rightarrow 0$), the more accurate approximation we are able to obtain ($\alpha, \delta \rightarrow 0$).

---

### Official Review · Reviewer_Ccvk · 2022-07-14

**Rating:** 5
**Confidence:** 4
**Soundness:** 3 good
**Presentation:** 2 fair
**Contribution:** 3 good

**Summary:**

This paper considers the problem of maximizing a monotone submodular function under matroid constraints when the value queries have independent random noise drawn from generalized exponential tail distributions. The first result is for uniform matroid constraints (aka cardinality constraints). Here, the new result improves on existing work in that the cardinality range over which it applies is larger than that of previous work while preserving polynomial dependence of query complexity on the approximation parameter $\epsilon$. Prior work either applied only to larger cardinality values $\Omega(\log \log n/\epsilon^2)$ against $\Omega(1/\epsilon)$ or required exponential dependence of query complexity on $\epsilon$. The second result is for general matroid constraints and obtains a constant approximation. Prior work had not considered general matroid constraints in maximizing submodular functions in the noisy setting.

The main technique is to use local search where each step swaps a single element. Prior work for noisy submodular maximization mostly used variants of the greedy algorithm. The local search algorithm is an adaptation of prior work by Filmus and Ward for exact queries, and the optimization function used for local search is also a robust version of the function used by FW.

**Questions:**

I do not have specific questions.

**Limitations:**

The paper includes a discussion of limitations at the end, but it seems largely perfunctory. As I mentioned above, it would be interesting to have a longer discussion about the noise model with its application scenarios and limitations, for instance. Also, are the cardinality bounds in this paper tight? If not, what are the challenges in removing this constraint altogether? Discussions of this sort would help make the paper a more engaging read.

**Strengths And Weaknesses:**

Strengths:

1. Submodular function maximization is an important classical problem with applications spanning many areas.

2. The paper gives the first results for submodular maximization under general matroid constraints in the noisy setting. Without noise, this is a standard setting in which submodular maximization is studied, so this is a good research direction in general. The approximation constant doesn't appear very impressive, but as the first paper considering this problem, it provides a good start.

3. The adaptation of the local search technique to the noisy setting might be useful in general. Local search is used in many contexts and constructing a robust version in the presence of noise might have value beyond the results in this paper.

Weaknesses:

1. The techniques employed in this paper are not very novel. The adaptation of the FW algorithm is fairly straightforward, and as such, the paper does not contribute much in terms of new techniques.

2. The noise model comes across as quite limited, and the paper does not include a discussion on it. It is true that a couple of previous papers also considered the same noise model, but I feel it would be useful to discuss the contexts in which this type of noise distribution is relevant.

---

> ### Author Response · Authors · 2022-08-02
> **Re: Official Review of Paper4250 by Reviewer Ccvk**
>
> Thanks for your comments that give us the chance to clarify our novelty and contributions. We hope that our reply addresses your concerns and that you will increase your support for the paper.
>
> > ''The adaptation of the FW algorithm is fairly straightforward, and as such, the paper does not contribute much in terms of new techniques.''
>
> While we agree that we use some ideas from [5], novel technical ideas are required in our paper. Our unified framework (Algorithm 1) is a generalization of [5] in the presence of noise, via a novel idea of introducing surrogate functions $h$ (Definition 3.2). The design of $h$ is non-trivial and needs to meet two properties: 1) $h$ should depend on an averaging set (Line 232) of size $\mathrm{poly}(n)$, such that $h(S)$ and its noisy analogue $\tilde{h}(S)$ are close for all $S$ considered by the FW algorithm; 2) $h$ needs to be close to the original function $f$, such that optimizing $h$ can yield a near-optimal solution for optimizing $f$. However, a large averaging set is more likely to induce a large gap between $h$ and $f$, making simultaneous fulfillment of both properties non-trivial. For example, a straightforward idea might be letting $h(S) = \frac{1}{2^{|S|}} \sum_{T \subseteq S} f(T)$, i.e., the expectation over all subsets of $S$. However, this surrogate satisfies neither of the properties mentioned above.
>
> Another technical idea is the $\tilde{f}$-maximization phase applied in Algorithm 2 and 3. We take Algorithm 2 as an example. A natural idea is to simply apply local search (Algorithm 1) to optimize $h(S) = \frac{1}{n} \sum_{e\in N} f(S+e)$ (Definition 4.2) under $r$-cardinality constraint. However, this does not yield a provable approximation of $max_{S:|S|\leq r} f(S)$, since it is not easy to control the contribution of the additional element $e\in N$ to $h(S)$ and it is possible that $h(S)\gg f(S)$. To handle this difficulty, Algorithm 2 first runs a local search procedure under $(r−1)$-cardinality constraint and then selects an additional element with a ''large enough'' margin at the $\tilde{f}$-maximization phase. This idea enables us to control the loss induced by the surrogate $h$ within $\frac{1}{r}\cdot OPT$.
>
> > ''The noise model comes across as quite limited, and the paper does not include a discussion on it. It is true that a couple of previous papers also considered the same noise model, but I feel it would be useful to discuss the contexts in which this type of noise distribution is relevant.''
>
> Thank you for the valuable suggestion. There are many models in the literature where set functions are distorted by random variables drawn i.i.d from some distribution. Reference [18] has included some examples such as revealed preference theory [CE16] and active learning [Fel09].
>
> As for non-i.i.d. distributions, Reference [10] indicates that no algorithm can achieve a constant approximation when the noise multipliers are arbitrarily correlated across sets (Lines 33-35). Considering this, it may be worthwhile to consider special cases for which optimal guarantees can be obtained. One of them is a model called $d$-correlated noise [9]: a noise distribution is $d$-correlated if for any two sets $S$ and $T$ such that $|S\backslash T|+|T\backslash S| > d$, the noise is applied independently to $S$ and $T$. Otherwise, the noise multipliers can be arbitrarily correlated. We notice that our algorithms can be naturally extended to $d$-correlated noise for $d=O(1)$. In particular, to adapt Algorithm 2 to deal with $d$-correlated noise, we need to arbitrarily split $N$ into sets $T_1,\dots,T_{L}$ ($L=\lfloor\frac{N}{d+1}\rfloor$) and define the smoothing surrogate function as $h(S) = \frac{1}{L}\sum^L_{l=1}f(S\cup T_l)$ to replace the original one.
>
> We will add this discussion in the future version.
>
> > ''Are the cardinality bounds in this paper tight? If not, what are the challenges in removing this constraint altogether?''
>
> For any cardinality $r\geq 2$, our algorithms return a $(1-\frac{1}{r})(1-\frac{1}{e})-O(\epsilon)$-approximation with query complexity $\mathrm{Poly}(n, 1/\epsilon)$. When $r\in \Omega(1/\epsilon)$, the approximation ratio can be written as $\left(1-\frac{1}{e}\right)-O(\epsilon)$ (Theorem 1.1). However, if $r\in O(1/\epsilon)$, the approximation ratio $(1-\frac{1}{r})(1-\frac{1}{e})-O(\epsilon)$ is not optimal. In this case, [18] provides an algorithm that achieves $\left(1-\frac{1}{r}\right)$-approximation with query complexity $\Omega(n^r)$. It remains open whether there is an algorithm that returns $\left(1-\frac{1}{e}-O(\epsilon)\right)$-approximation with query complexity $\mathrm{Poly}(n, 1/\epsilon)$ for $r\in O(1/\epsilon)$. The main challenges of our approach to handle this range of $r$ is that the introduction of surrogates (Definition 4.2) inevitably results in a loss of $1/r$ in approxiamtion ratio (Line 262).
>
> We will add this discussion to the conclusion section in the future version.

---

### Meta-Review · Area_Chair_kRfn · 2022-08-24

**Recommendation:** Accept
**Confidence:** Certain

**Metareview:**

Overall, this paper studies an important problem and obtains strong results. There were some concerns raised by one reviewer regarding the novelty of the techniques and the lack of discussion regarding the noise model. Please make sure to add the discussion about the noise model from the rebuttal in the camera ready version of the paper.

**Award:**

No

---

### Decision · Program_Chairs · 2022-09-14

Accept